# A HaloTag-TEV genetic cassette for mechanical phenotyping of proteins from tissues

Jaime Andrés Rivas-Pardo [1,2,6], Yong Li[3,6], Zsolt Mártonfalvi [4], Rafael Tapia-Rojo[1], Andreas Unger[3], Ángel Fernández-Trasancos[5], Elías Herrero-Galán[5], Diana Velázquez-Carreras [5], Julio M. Fernández[1], Wolfgang A. Linke [3✉] & Jorge Alegre-Cebollada [5✉]

Single-molecule methods using recombinant proteins have generated transformative hypotheses on how mechanical forces are generated and sensed in biological tissues. However, testing these mechanical hypotheses on proteins in their natural environment remains inaccesible to conventional tools. To address this limitation, here we demonstrate a mouse model carrying a HaloTag-TEV insertion in the protein titin, the main determinant of myocyte stiffness. Using our system, we specifically sever titin by digestion with TEV protease, and find that the response of muscle fibers to length changes requires mechanical transduction through titin's intact polypeptide chain. In addition, HaloTag-based covalent tethering enables examination of titin dynamics under force using magnetic tweezers. At pulling forces < 10 pN, titin domains are recruited to the unfolded state, and produce 41.5 zJ mechanical work during refolding. Insertion of the HaloTag-TEV cassette in mechanical proteins opens opportunities to explore the molecular basis of cellular force generation, mechanosensing and mechanotransduction.

[1] Department of Biological Sciences, Columbia University, New York, NY 10027, USA. [2] Center for Genomics and Bioinformatics, Facultad de Ciencias, Universidad Mayor, Santiago, Chile. [3] Institute of Physiology II, University of Muenster, Muenster, Germany. [4] Department of Biophysics and Radiation Biology, Semmelweis University, Budapest, Hungary. [5] Centro Nacional de Investigaciones Cardiovasculares (CNIC), Madrid, Spain. [6] These authors contributed equally: Jaime Andrés Rivas-Pardo, Yong Li. ✉email: wlinke@uni-muenster.de; jalegre@cnic.es

The behavior of cells is decisively regulated by mechanical cues, which are generated, sensed and transduced by specialized proteins[1,2]. For example, integrin and talin transduce mechanical signals from the extracellular matrix into the cell through elastic structural domains that unfold and refold under force, leading to modulation of the binding affinity for downstream effectors[3,4]. The interplay between mechanical force and biology is even more apparent in cell types whose primary function is force generation, such as myocytes. These cells enable contraction of cardiac and skeletal muscle thanks to the activity of sarcomeres, which are highly organized protein structures that ensure efficient mechanical power delivery from arrays of actomyosin filaments[5]. The giant protein titin is a fundamental component of sarcomeres, where it is continuously subject to end-to-end pulling force (Fig. 1a). Spanning half the length of a sarcomere, titin sets the passive stiffness of myocytes and enables force transduction and coordination between sarcomeres[6]. As a result, titin is essential for the mechanical activity of striated muscle. Indeed, mutations in the titin gene are a major cause of heart disease and myopathies[7–9].

Single-molecule force-spectroscopy methods, in particular Atomic Force Microscopy (AFM), have been extensively used to examine the response to force of proteins with mechanical functions, including titin[10,11]. These experiments have shown that proteins, when placed under a pulling force, unfold and refold according to their underlying free energy landscapes, which are highly force dependent[12,13]. The fact that polypeptides are able to fold under load implies that they can generate mechanical work[13,14]. Indeed, the magnitude of mechanical work produced by titin folding has been proposed to be similar to that delivered by the ATP-driven activity of myosin motors, a realization that has triggered the hypothesis that titin folding is an important contributor to active muscle contraction[15,16]. However, hypotheses like this are challenging to test experimentally due to the absence of methods to probe native protein mechanics.

Single-molecule methods involve purification of short, engineered recombinant proteins, which are exposed to pulling forces that can be far from the physiological range[17]. To obtain functional insights, extrapolating models take into account physiological ranges of forces and actual protein sizes[10]. However, these models are based on approximations whose validity is difficult to assess. Recent technological breakthroughs address some of these limitations. For instance, optical tweezers and magnetic-tweezers (MT)-based force spectroscopy enable single-molecule measurements directly at physiological ranges of force[18–20]. In addition, MT measurements are done under exquisite control of force (force-clamp) and are stable over long time scales[21]. Still, the study of intact elastic proteins by single-molecule methods is limited due to the size of many of these proteins, ranging from 200 to 3700 kDa[22–24], which is challenging to reach by heterologous protein expression systems[10,25–27]. In addition, recombinant proteins do not contain native posttranslational modifications, which are key factors modulating the mechanical properties of proteins[28]. Although proteins like titin can be purified from some natural sources and probed mechanically[11,29,30], not every tissue is suited for analysis due to limited purification yield. More importantly, controlling the pulling geometry in native proteins is not possible, which leads to ambiguous assignment of mechanical features[11,20].

In this study, we develop a knock-in genetic cassette that enables controlled examination of the mechanical properties of homologously produced proteins. Our cassette includes a TEV protease recognition site and a HaloTag domain, which in combination allow protein labeling to assess cellular localization, polypeptide severing to probe mechanical function, and click-chemistry-based covalent anchoring for single-protein mechanical

manipulation. We have used the HaloTag-TEV cassette to study titin (Fig. 1a). Our results confirm that forces <10 pN readily recruit titin domains to the unfolded state in a reversible manner[20]. We also demonstrate that the mechanical folding of titin is modulated across a wider range of forces than previously estimated using short, recombinant titin fragments. By specifically digesting HaloTag-TEV titin with TEV protease, we have been able to quantify the contribution of titin to the overall stiffness of muscle fibers. We propose that the HaloTag-TEV cassette can be applied to examine the performance of a broad spectrum of proteins with mechanical function.

## Results

**Insertion of the HaloTag-TEV cassette in titin.** The I-band section of titin, containing up to 100 immunoglobulin-like (Ig) domains, is a major contributor to the passive stiffness of striated muscle tissue[22,31,32]. This segment includes an alternatively spliced region that provides titin with muscle-specific, tailored mechanical properties[6,31,33]. To be able to study the mechanical role of titin's I-band in all striated muscles, we have introduced the HaloTag-TEV cassette in-frame between constitutively expressed exons 225 and 226. These exons code for Ig domains I86 (residues 14072–14157, UniProt A2ASS6-1) and I87 (residues 14161–14246) (Fig. 1a, Supplementary Fig. 1)[6], which are located at the end of the I-band of titin. Homozygous and heterozygous animals for the HaloTag-TEV insertion are obtained at Mendelian rates (heterozygous crossing resulted in 24% wild-type, 49% heterozygous, and 27% homozygous mice, $n = 88$). Both heterozygous and homozygous HaloTag-TEV-titin mice are fertile and appear as healthy as wild-type littermates. Indeed, homozygous mice have normal body weight (Supplementary Fig. 2A) and do not show any alterations in serum markers of inflammation or muscle damage (Supplementary Fig. 2B). In addition, the HaloTag-TEV-titin mice have normal heart function and resistance to exercise (Supplementary Fig. 2C–E). We did not observe any alteration in the ultrastructure of sarcomeres, their resting sarcomere length or in the passive force generated by muscle fibers from the HaloTag-TEV-titin mice (Supplementary Fig. 3). Overall, these results indicate that homozygous HaloTag-TEV-titin mice are healthy and that the insertion of the HaloTag-TEV cassette in titin does not perturb the ultrastructure or the mechanical function of sarcomeres.

The HaloTag domain enables click-chemistry reactions with chloroalkane ligands[34]. We took advantage of this functionality to examine the expression and localization of the HaloTag-TEV construct in homozygous mice. We first incubated gastrocnemius muscle with Oregon Green HaloTag ligand, and visualized the sample by multiphoton, confocal and STED fluorescence microscopies after fixation and clarification of the tissue. We detected intense labeling in doublet stripes that are transversally oriented with respect to the long axis of the fibers in agreement with the engineered location of the HaloTag-TEV cassette (Fig. 1a, b, Supplementary Movie 1). To further characterize the position of the HaloTag-TEV cassette in the sarcomere, we isolated bundles of demembranated myofibrils from psoas muscle and co-stained them with Alexa488 HaloTag ligand and Alexa 647-phalloidin (Fig. 1c). Since phalloidin stains only naked actin, it can be used to determine the location of the Z-disk and the pointed ends of the thin filaments[35]. As expected, the Z-disk, which is identified by the strongest phalloidin staining, appears between Alexa 488 doublets (Fig. 1c). Equivalent results were obtained by observing sarcomeres directly under brightfield illumination using a spinning disk microscope (Supplementary Fig. 4). Distances between fluorescent stripes were estimated from experiments with single myofibrils (Fig. 1d). While the long distance between Alexa 488 stripes was always 1.5–1.6 μm, in

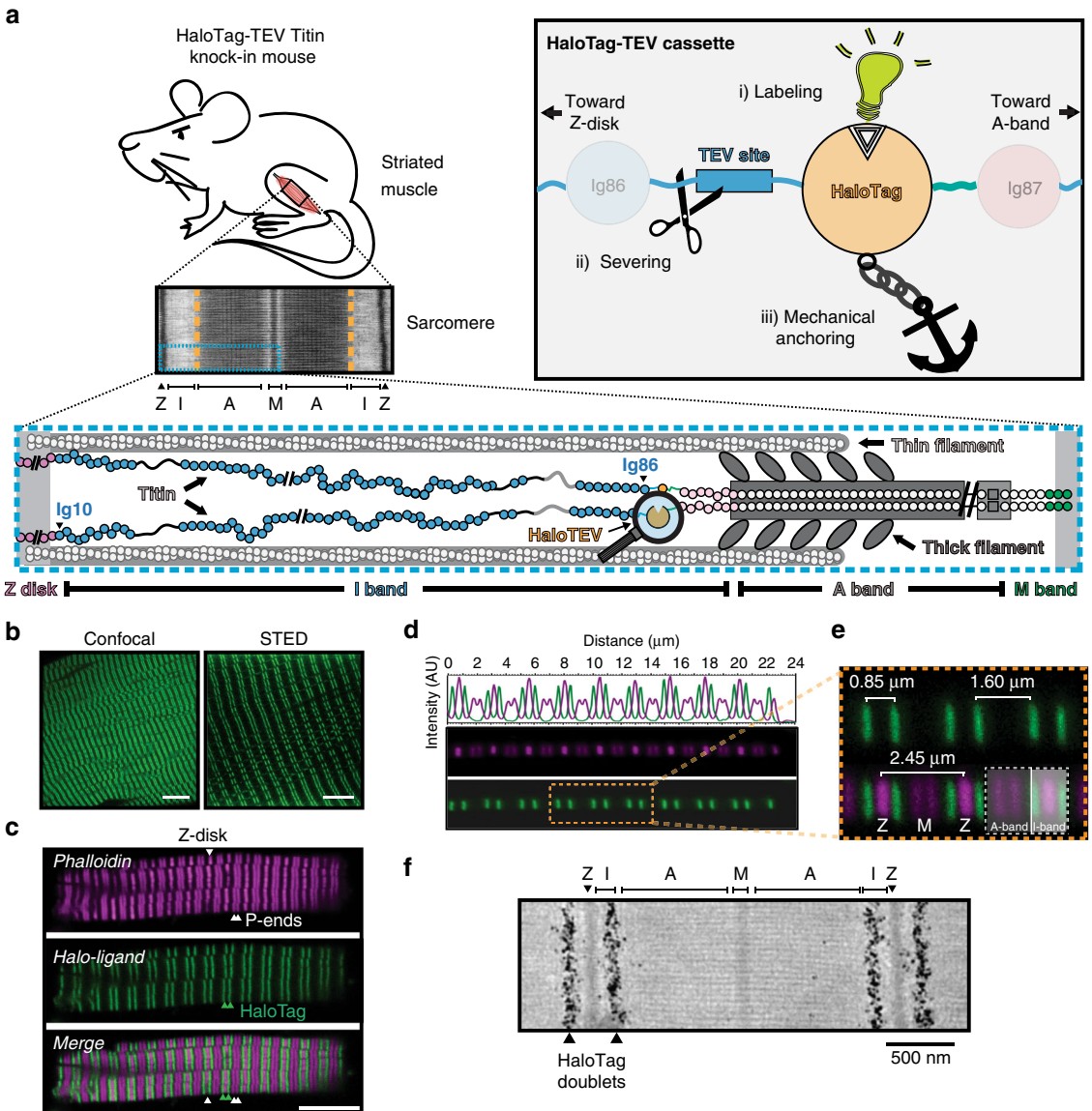

**Fig. 1 The HaloTag-TEV cassette. a** We inserted the HaloTag-TEV cassette in the *TTN* gene to generate a mouse model for the study of the mechanical properties of titin. Bottom: Schematic representation of one half-sarcomere showing the three main filaments of sarcomeres: the thin filament, the thick filament and titin. A single-titin molecule spans from the Z-disk to the M-band. The HaloTag-TEV knock-in cassette is located at the end of the I-band, between the I86 and I87 domains. Inset: The HaloTag-TEV cassette allows (i) specific labeling, (ii) titin severing to examine tissue mechanics, and (iii) covalent anchoring for single-molecule force spectroscopy. **b** Gastrocnemius muscle extracted from homozygous knock-in HaloTag-TEV titin mice is labeled with the membrane permeable Oregon Green HaloTag ligand, fixated and cleared following the X-Clarity method. The specimen is then imaged using confocal and STED microscopies. The scale bars correspond 10 and 5 μm, respectively (n = 1 experiment; more than five fields of view showed equivalent results). **c** Freshly extracted psoas myofibrils are stained with Alexa488 HaloTag ligand and Alexa Fluor 647 phalloidin, and subsequently imaged using confocal microscopy (experiment was repeted three times with two different muscles leading to equivalent results). Phalloidin specifically binds naked actin, resulting in strong fluorescent signal at the Z-disk region and the pointed ends of the thin filament (P-ends), as indicated by white arrowheads. According to the pattern of phalloidin labeling, the HaloTag appears correctly positioned flanking the A-band of every sarcomere (green arrowheads). Thus, the HaloTag doublet pattern corresponds to the HaloTag-TEV cassette flanking two I-bands from two juxtaposed sarcomeres. Scale bar, 7.5 μm. **d**, **e** A single psoas myofibril labeled with Alexa488 HaloTag ligand and Alexa Fluor 647 phalloidin is used to measure the distance between fluorescent bands. The HaloTag doublets are separated approximately 1.5–1.6 μm, while the distance between the bands in the doublets is variable (0.85 μm in this particular example) as a consequence of different contraction states of sarcomeres (representative image from four preparations). **f** Representative EM image out of n = 200 micrographs showing the location of gold-particle-labeled HaloTag in soleus muscle.

agreement with the constant length of murine A-band, the distances between Z-disks were quite variable reflecting the different contraction states of individual sarcomeres in rigor buffer (Fig. 1e)[36]. We further verified the position of the HaloTag domain at the end of the I-band by nanogold electron microscopy (EM) using biotinylated halo-ligand, which was detected with gold-streptavidin (Fig. 1f).

Taken together, our results show that the insertion of the HaloTag-TEV cassette in the titin gene is well tolerated in the mouse, and results in a functional HaloTag domain that localizes

to the expected region of the sarcomere, both in skeletal and in cardiac tissue (Supplementary Fig. 5).

**TEV-mediated severing hinders titin mechanical function.** The contribution of titin to the stiffness of muscle is usually considered to be prominent. This view is supported by experiments where harsh extraction methods are used to deplete titin from myocytes[31,37]. However, the specificity of these methods is limited since other cytoskeletal components can be affected, which can result in overestimation of the contribution of titin to muscle stiffness. The HaloTag-TEV cassette offers the opportunity to use the highly specific TEV protease to cleave a single peptide bond in titin (Fig. 2a), resulting in cessation of force transduction through titin's polypeptide chain while preserving all other components in the myocyte.

To investigate whether specific severing of the titin filament is possible, we first purified a recombinant version of the HaloTag-TEV cassette flanked by Ig domains I86 and I87. The I86-HaloTag-TEV-I87 construct is readily digested by treatment with TEV protease (Supplementary Fig. 6), confirming accessibility of the TEV site. We then incubated cardiac muscle from wild-type (WT) or homozygous HaloTag-TEV mice with TEV protease, and analyzed protein content using low-percentage acrylamide SDS-PAGE and Coomassie staining. The resulting pattern of titin bands is not affected by incubation with TEV when WT muscles are probed (Fig. 2b). In these WT samples, low-electrophoretic-mobility bands corresponding to the N2BA and N2B titin isoforms (3.2–3.0 MDa) are identified along with T2 degradation products and/or the Cronos isoform of titin[31,38]. In contrast, TEV protease digestion of HaloTag-TEV muscle leads to the disappearance of the N2B and N2BA bands, which are replaced by three bands whose mobilities are in agreement with the expected 2.2 MDa size of titin's A-band/M-band region and the size of the Z-disk/I-band fragments originated from both the N2B and N2BA isoforms (Fig. 2b). TEV-digestion experiments with skeletal muscles containing the long N2A titin isoform showed equivalent results (Supplementary Figs. 7–9). As a further proof of specific digestion, we incubated the samples with HaloTag Alexa488 ligand before electrophoresis. The only fluorescent band after digestion corresponds to the A-/M-band region of titin, in agreement with the relative position of the TEV site and the HaloTag in the cassette (Supplementary Fig. 7B). Equivalent specific fluorescent labeling was also observed in digested I86-HaloTag-TEV-I87 recombinant protein (Supplementary Fig. 6). Western blot analyses using MIR antibody, which targets titin's A-band edges[39,40], further support specific digestion of HaloTag-TEV titin by TEV (Supplementary Fig. 7C).

Having demonstrated TEV-mediated specific severing of HaloTag-TEV titin, we measured the stiffness of TEV-treated skeletal myofiber bundles by subjecting them to step increases in length and monitoring the resulting passive force. Treatment of HaloTag-TEV psoas myofibers with TEV resulted in a time-dependent drop in passive force of ~50% in conditions in which cleavage of titin was complete (Fig. 2c, d, Supplementary Figs. 7D, 8). This softening effect did not occur when WT myofibers were incubated with TEV, ruling out the potential contribution of off-target TEV sites (Fig. 2c, d). Further supporting the absence of relevant off-target sites, we did not find differences in the pattern of bands obtained in 12.5% SDS-PAGE following TEV treatment of muscle fibers, in conditions in which titin was readily cleaved (Supplementary Fig. 9). In these mechanical experiments, there was some variability in the time required for TEV to achieve full digestion, probably reflecting lack of control of the diameter of the specimens and the degree of skinning, which may influence accessibility of TEV protease to cleavage sites. When digestion

was complete, the drop in passive tension was consistently around 50% (Fig. 2d). Traditional experiments aimed at dissecting the contribution of titin to overall passive muscle stiffness from that of other structural components have used high ionic strength KCl/KI extraction buffer. Addition of this extraction buffer to our TEV-treated HaloTag-TEV psoas myofibers at the end of the mechanical protocol resulted in an additional decrease in passive tension (Fig. 2d), although titin was confirmed (by gel electrophoresis) to be fully cleaved in these experiments. When we studied the structural preservation of TEV-treated HaloTag-TEV myofibers at the end of stepwise stretch-release protocols by electron microscopy, we found the sarcomere ultrastructure to be severely compromised; sarcomere proteins sometimes appeared lumped together in an irregular manner (Supplementary Fig. 10A–C).

Since the Ig domains close to the HaloTag-TEV cassette can multimerize in muscle sarcomeres[41], potentially leading to alternative multi-titin force transmission pathways, we repeated mechanical testing experiments in which the TEV protease is added at long sarcomere lengths. Under these conditions, the titin filament is extended and multimerization of domains from opposite sides of the cleavage site is presumably minimized. On electron micrographs, the stretched, TEV-treated HaloTag-TEV myofibers showed wavy Z-disks and A-bands but the overall regularity of the sarcomere structure was better preserved than in myofibers undergoing cycles of stretch-release (Supplementary Fig. 10D). In the mechanical experiments in which the TEV protease is added at long sarcomere lengths, we found a ~65% reduction of passive force (Fig. 2e), consistently higher than the value obtained when TEV protease is added to slack fibers that were then subject to cycles of stretching and release (Fig. 2d). These results suggest that there could be out-of-register multimerization of titin domains or other non-specific protein–protein interactions generated by the cleavage of titin, which, however, can be minimized if the TEV-treated HaloTag-TEV-titin myofibers are held in the stretched state. Therefore, the ~65% reduction of passive stiffness measured upon titin cleavage in the stretched state presumably reflects the contribution of titin to overall passive stiffness better than the ~50% reduction measured in the stepwise stretch-release protocols.

**Directed manipulation of single-titin molecules.** The HaloTag domain has been used to covalently link short recombinant proteins to glass surfaces for single-molecule force-spectroscopy experiments[42]. We have extended this technology to titin molecules extracted from HaloTag-TEV titin knock-in mice, and used recently developed high-resolution single-molecule MT to pull specifically the elastic I-band part of titin (Fig. 3). To this end, we grabbed HaloTag-immobilized titin using paramagnetic beads coated with T12 antibody, which recognizes the first few proximal Ig domains in titin's I-band and has been validated in optical tweezers experiments[20,43,44] (Fig. 3d). In our MT setup, a pair of permanent magnets are brought in close proximity to the sample. As a result, a net constant magnetic pulling force is generated on anchored molecules. The resulting length of the tethered titin molecule is measured by comparing the position of the paramagnetic bead and a reference bead glued to the glass surface[18].

Three typical MT pulling recordings coming from single-titin molecules isolated from different muscles are shown in Fig. 3e. These traces, which are obtained at a final force of 19.7 pN, feature step increments in the length of the proteins that originate from mechanical unfolding of titin domains[20,45]. We measured the extension of the unfolding events at this pulling force, finding two major populations of step sizes at $21 \pm 6$ and $7 \pm 3$ nm (mean $\pm$ SD of the Gaussian distributions, Fig. 3e). An extension

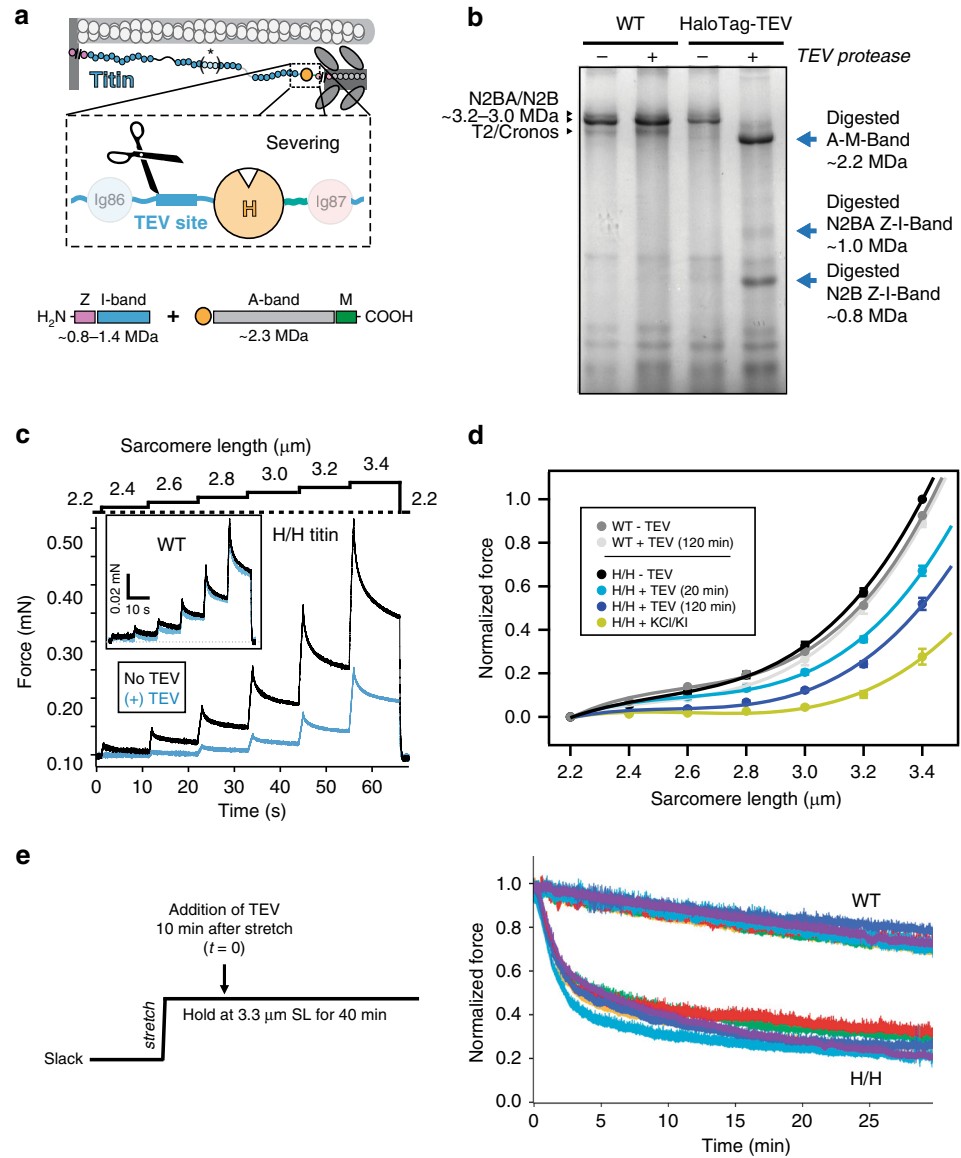

**Fig. 2 Directed severing of HaloTag-TEV titin. a** Following treatment with TEV protease, the HaloTag-TEV titin is expected to result in at least two fragments: A-M-band fragment (~2.2 MDa) and the Z-disk/I-band fragment(s), which can show a range of molecular weights depending on the specific titin isoform (represented by asterisk and brackets in the titin diagram). **b** Digestion of cardiac titin is tracked by 1.8% polyacrylamide SDS-PAGE electrophoresis and Coomassie staining (a representative image of 20 experiments showing the same result). Wild-type titin is not affected by incubation with TEV protease, so typical bands corresponding to full size N2BA/N2B isoforms and to the T2 fragment or the Cronos isoform are observed in both lanes. In contrast, digestion of HaloTag-TEV titin results in a main band at 2.2 MDa, corresponding to the A-M-band fragment, and two extra bands at 0.8 and 1.0 MDa that are assigned to the I-band portions of the N2B and N2BA cardiac isoforms, respectively. **c** Passive stiffness of skinned bundles of psoas myofibers isolated from homozygous HaloTag-TEV titin mice (H/H). During mechanical testing, myofibers were stretched in steps to increasing sarcomere lengths, and the resulting passive force was recorded. Force was first measured in relaxing buffer, then myofibers were incubated with TEV for 30 min at slack length (2.2 μm), and the measurement was repeated. Inset: control experiment with wild-type fibers. **d** The drop in passive force that results from incubation of HaloTag-TEV titin fibers with TEV protease is time-dependent (in these experiments, H/H fibers were returned to slack length between consecutive mechanical tests at 10, 20, 40, and 120 min; data at 10 and 40 min are not presented for clarity). After 120 min, high ionic strength KCl/KI buffer was added to extract titin. To average several experiments (n = 6), force values are normalized to the value at 3.4 μm SL before treatment with TEV. Error bars represent SEM. Source data are provided as a Source Data file. **e** Mechanical testing at long sarcomere lengths. Left: Experimental protocol. Right: Results from six independent experiments using ~150-μm-diameter skinned psoas fibers.

of 21 nm is compatible with full mechanical unfolding of Ig domains[11,45]. The shorter unfolding steps represent ~12% of the total events detected in mechanical unfolding trajectories. Since the population of short step sizes is well defined, we consider it highly unlikely that the short steps are due to spurious interactions. Similar shorter step sizes have been detected before when pulling from recombinant proteins if domains contain disulfide bonds[46,47], if there are unfolding intermediates[48–50], or if a population of molten globule structures is present[51]. Any of these mechanisms could contribute to explain our observations pulling from titin's I-band. It is also possible that the short steps are due to yet-to-characterize unfolding elements within the

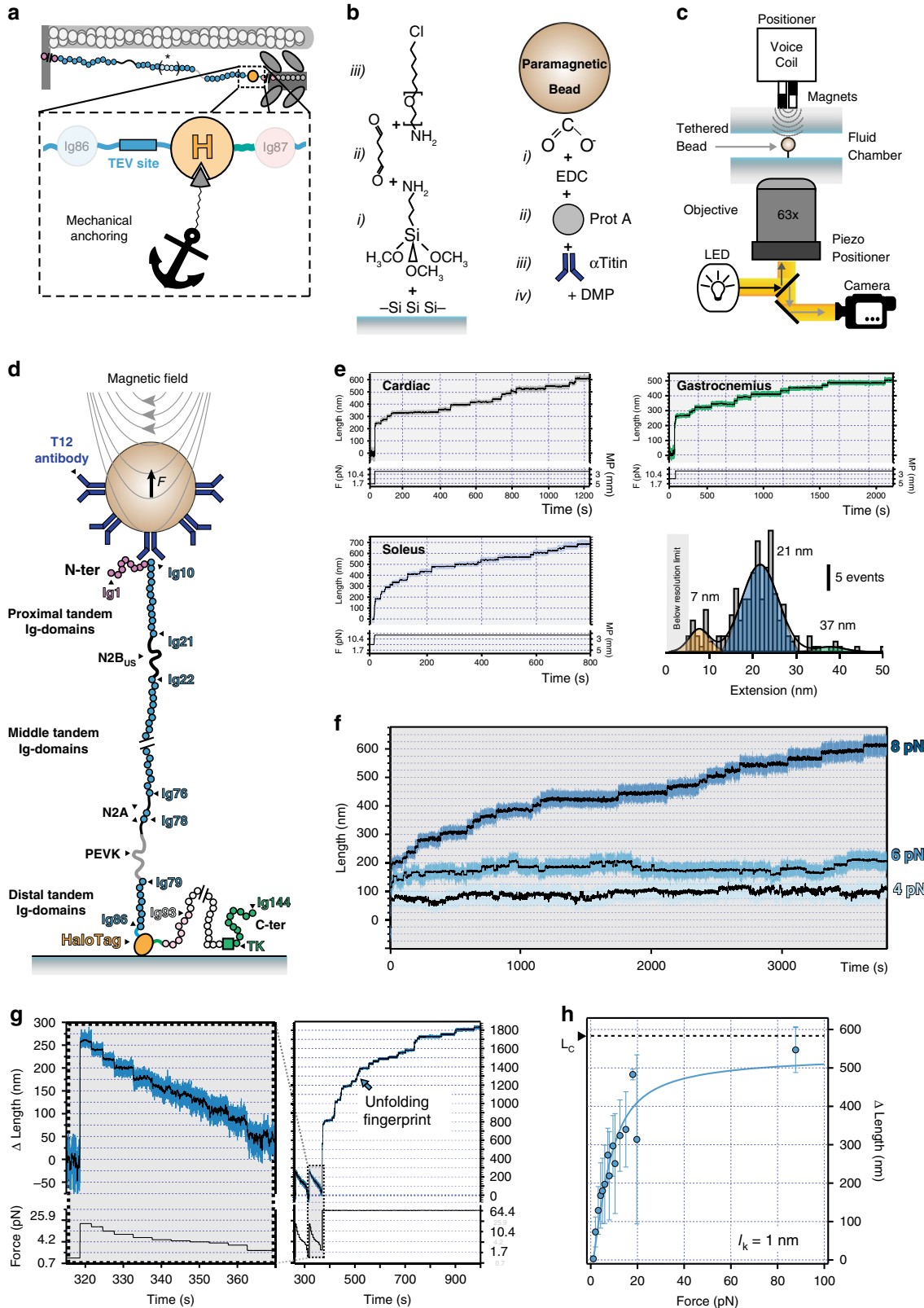

N2B$_{US}$, the N2A's unique elements and the PEVK. We also detected occasional steps whose length is longer than theoretically allowed for single Ig domains. We interpret these steps as originating from two or more misfolded domains that exchange β-strands to complete the folded state[52–54]. We measured how the size of the unfolding steps varies with force and found the typical scaling predicted by models of polymer elasticity. Using the worm-like chain model[55], we estimate that the contour length increments ($\Delta L_C$) of the main unfolding populations of steps are 30 ± 1 nm and 14 ± 1 nm (Supplementary Fig. 11).

**Fig. 3 Magnetic-tweezers-based single-molecule force spectroscopy of HaloTag-TEV titin. a** Scheme of HaloTag-mediated titin immobilization. **b** Glass and bead functionalization by surface chemistry (see Methods for details). **c** Cartoon showing the MT setup[15,18]. **d** Using T12-antibody-functionalized paramagnetic beads and HaloTag-derivatized glass surfaces, only the elastic segment titin I-band is stretched during MT pulling experiments. Relevant regions of the N2BA isoform of titin are indicated (nomenclature of domains according to Uniprot A2ASS6-1). **e** Titin molecules purified from three different muscles are stretched at a final force 19.7 pN, resulting in unfolding steps. The histogram shows three different populations of step sizes. Initial forces were 1.7 pN (cardiac) and 4.2 pN (gastrocnemius and soleus). **f** Naive titin molecules isolated from gastrocnemius muscle are pulled at 4–8 pN, which results in recruitment of domains to the unfolded state. **g** Left: The length of naive gastrocnemius titin tethers before domain unfolding is measured at different forces and referenced to the length measured at 1.1 pN to obtain Δ Length. Right: tethers are then subject to high force to verify single-molecule tethering from the fingerprint of domain unfolding steps. **h** Δ Length estimated at different forces for gastrocnemius HaloTag-TEV titin tethers ($n = 244$ independent measurements from five independent molecules). Data are presented as mean values ± SD. Solid line is a fit to the Freely Jointed Chain model of polymer elasticity; the resulting Kuhn length is shown in the inset. Source data are provided as a Source Data file.

All titin domains, including those belonging to the mechanically silent A-band portion, unfold when placed under mechanical force[20]. A specific feature of titin's I-band region is the presence of the PEVK and $N2B_{US}$ random coil regions (Fig. 3d), which are characterized by low persistence/Kuhn lengths (typical Kuhn length, $l_k$, of <2 nm versus 20 nm for stretches of folded domains[10]). Hence, to show that our experiments are indeed probing specifically the I-band segment of titin, we estimated $l_k$ of our single-titin tethers. We measured the initial length before any unfolding event of gastrocnemius HaloTag-TEV titin at different forces. In these experiments, we subjected titin molecules to short pulses of forces between 1.1 and 87.7 pN, and then pulled to high forces to confirm formation of single-molecule tethers from the appearance of unfolding steps (Fig. 3g). To avoid any preconception regarding the architecture of the tethers, we fitted the data to a single-term freely jointed chain model of polymer elasticity[56]. We obtained $l_k = 1$ nm (Fig. 3h), a low value compatible with the presence of random coil regions. Hence, the analysis of the initial lengths of single-protein tethers confirms specificity of HaloTag-mediated anchoring of the I-band region of titin. The obtained $l_k$ is a convolution of the values for both random coil regions and folded domains in gastrocnemius titin, so it can be seen as an upper limit for the $l_k$ of the PEVK region. This value is in the lower range of previous estimates using recombinant proteins[10]. Interestingly, phosphorylation has been shown to decrease the persistence length of the PEVK segment[57], raising the possibility that equivalent posttranslational modifications are present in titin isolated from muscle.

**Titin unfolding/folding transitions generate mechanical work.**
Taking advantage of the high stability and resolution of MT at low forces, we pulled the I-band of native HaloTag-TEV titin at forces lower than 10 pN. These experiments confirmed that these low forces are sufficient to recruit several Ig domains to the unfolded state in minutes (Fig. 3f)[20]. We also observed step decreases in length, which are caused by Ig-domain refolding events[15,20].

Folding contractions against a pulling force generate mechanical work, which has been quantified before using short titin fragments[15]. To inspect the range of forces at which titin folding operates, we used triple force pulse experiments (Fig. 4a). In these experiments, an initial fingerprint pulse to high forces rapidly recruits Ig domains to the unfolded state. During the refolding pulse at low forces, folding contractions occur following elastic recoil of the polypeptide. A final probe pulse is used to quantify the number of domains that managed to refold in the refolding pulse. In the trace in Fig. 4a, 20 unfolding steps are recorded in the probe pulse (arrowheads). Folding of 20 Ig domains is expected to cause a 200-nm folding contraction at 5 pN since every unfolding domain contributes ~10 nm (Supplementary Fig. 11B). This ~200 nm predicted contraction matches the

observed folding contraction (Fig. 4a), showing the robustness of the method.

To calculate the probability of folding at different forces, we set the force of the refolding pulse between 1 and 16 pN, and measured the ratio of unfolding domains observed in the fingerprint pulse vs those observed in the probe pulse. Long probe pulses over 300 s were used to ensure equilibrium. Compared with engineered polyproteins[15], the dependence of titin's I-band folding with force is less cooperative (Fig. 4b). Full folding only occurs at 1 pN, but some domains are able to fold against larger forces of up to 10 pN. The force at which the probability of folding is 0.5 ($P_{F0.5}$) is 6.5 pN, very similar to the value obtained using engineered polyprotein constructs[15]. We calculated the work delivered by folding considering the size of the refolding step at different pulling forces. We found that full-length Ig-domain folding can generate more than 100 zJ mechanical work at the highest forces (~10 pN, Fig. 4c). However, at these high forces, the folding probability is very low. By multiplying the work produced by folding by the folding probability, we obtain a more useful measurement of the average energy produced by the folding of a single-titin domain, which we found to be 41.5 zJ[15].

**Discussion**
We have developed a HaloTag-TEV genetic cassette to enable mechanical manipulation of targeted proteins from tissues. The cassette, which does not induce any striking phenotype when inserted in murine titin (Supplementary Figs. 2, 3), includes a HaloTag module and a recognition site for the highly specific TEV protease. While the TEV site can be exploited to halt force transduction via specific proteolysis (Fig. 2), the HaloTag-based covalent surface immobilization allows long attachment times required for in-depth protein mechanical characterization by single-molecule force spectroscopy (Figs. 3 and 4).

We have applied the HaloTag-TEV to probe the nano-mechanics of the I-band region of titin at forces below 20 pN. Previous approaches based on optical tweezers relied on non-specific tethering strategies[20,58], so the geometry of the single-molecule tethers could not be controlled. As a consequence, those milestone experiments also probed the mechanically silent A-band section of titin, precluding robust assignment of mechanical unfolding and folding transitions to the functionally relevant I-band region. Using HaloTag for directed single-molecule tethering, we have confirmed that at forces below 10 pN, a fraction of domains belonging to the I-band of titin are readily and reversibly recruited to the unfolding state (Fig. 3f)[10,20]. In this context, it will be interesting to explore whether recently described mechanical destabilization at low forces is a general feature of titin Ig domains[59], although the extent of titin domain unfolding in vivo is expected to depend on several additional factors including intra and extracellular constraints, the concomitant force relaxation

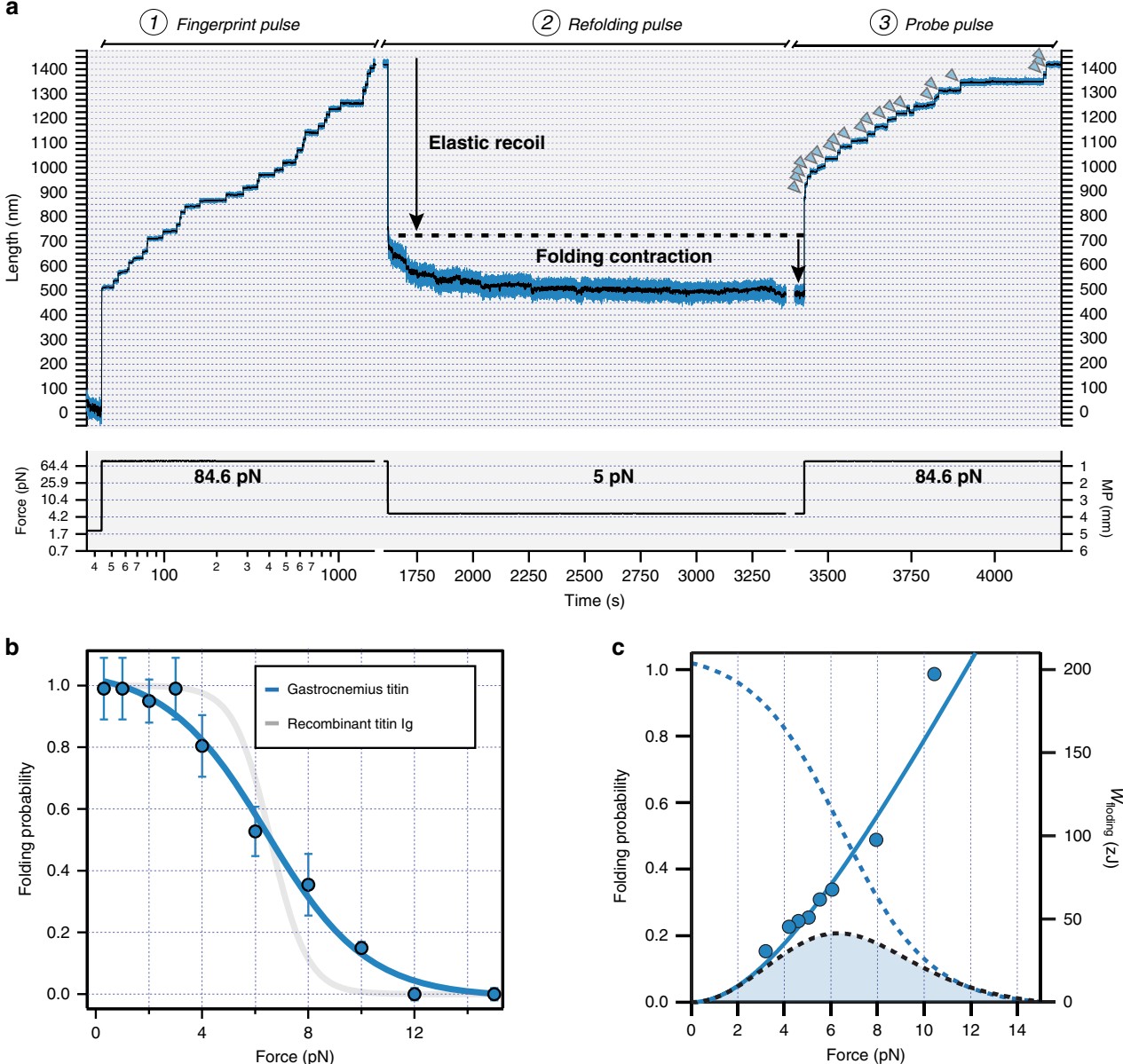

**Fig. 4 Mechanical work generated by titin folding. a** We follow a three-pulse force protocol. During the fingerprint pulse at high forces (1), Ig domains are recruited to the unfolded state. These domains are allowed to refold during the long refolding pulse at low force (2). To estimate folding fractions, we measure the number of domains that unfold in the final probe pulse (3), and compare with the initial number of unfolding domains in the fingerprint pulse. Arrowheads point to unfolding events in the probe pulse. The trace was obtained using gastrocnemius HaloTag-TEV titin. **b** Folding probability of the I-band of HaloTag-TEV titin isolated from gastrocnemius muscle (blue symbols). Solid line shows a sigmoidal fit to the data. The folding probability is less cooperative that the one obtained with engineered titin polyproteins (in gray, data from Rivas-Pardo et al.[15]), while the $P_{F0.5}$ remains very similar. Data are presented as mean values ± SD, $n = 120$ total measurements. Source data are provided as a Source Data file. **c** Force dependency of the work delivered by titin domain folding, which is obtained by multiplying step sizes by force (blue symbols). The solid line considers the step sizes obtained by the worm-like chain model. Considering the probability of folding (blue dashed line, data from panel **b**), we calculate the expected work delivered by titin folding (black dashed line, filled area), which peaks at 6.3 pN producing 41.5 zJ.

induced by unfolding, and the presence of other proteins in the sarcomere, such as chaperones.

To investigate the role of domain folding in active contraction, we have studied how the domains of I-band titin are able to refold against varying pulling forces. In our experiments, we have used long quenches at low forces to ensure equilibrium. We demonstrate that the force dependency of titin's folding is less cooperative than previously estimated using recombinant fragments (Fig. 4b). We propose that this low cooperativity in native titin's mechanical folding can stem from heterogeneous mechanical

properties of the multiple contributing Ig domains[10] and from mechanically active posttranslational modifications, such as disulfide bonds, S-thiolation and phosphorylation[28,60], which are challenging to incorporate in recombinant proteins. In any case, this low cooperativity comes with the advantage that titin can generate mechanical work at a wider range of forces than previously estimated[15]. Hence, our results suggest that during the lengthening phase of muscle activity, titin Ig domains can be recruited to the unfolded state to store elastic energy, which can be delivered as productive work during contraction. In this

scenario, the contribution of domain folding to contraction will depend on the physiological range of forces experienced by titin in sarcomeres, generally estimated to be below 10 pN[10,15,32,61].

Our MT experiments show that a reduction in force leads to titin contraction by two independent mechanisms, i.e., fast elastic recoiling due to the entropic adjustment of polymer length and Ig folding. In the example shown in Fig. 4a, the contribution of the elastic recoil predominates due to the abrupt change in force from 84.6 to 5 pN. However, the contribution of both factors is anticipated to be reverted in vivo, where changes in force acting on titin are calculated to be below 10 pN. Under such subtle changes in force, elastic recoiling is minimal; however, the folding ability of titin domains can vary dramatically (Fig. 4b). Hence, titin folding contractions emerge as key transitions that not only set the stiffness of titin, but also titin's ability to produce mechanical work. Indeed, we speculate that the work produced by folding domains can be largely regulated by muscle chaperones[62,63], molecular crowding[64], or posttranslational modifications[28,65,66]. The HaloTag-TEV titin can now be used to test these hypotheses.

Folding kinetics is a another key parameter that is expected to define the contribution of titin folding to active contraction, particularly in muscles operating at high frequencies (e.g., ~10 s$^{-1}$ in the case of the mouse heart)[67]. If folding is too slow with respect to muscle contraction, the energy released by folding will be minimal, if any. It has been recently observed that disulfide bonds can speed up Ig-domain folding to rates approaching 100 s$^{-1}$ at forces <10 pN[68]. To measure the speed at which native titin Ig domains are able to fold against a pulling force, new analysis methods will have to consider the presence of long random coil regions in titin. In this regard, magnetic tweezers setups with much improved temporal resolution can be key to capturing fast folding transitions of titin domains[51].

Force transduction through proteins requires covalent continuity throughout the polypeptide chain. We have demonstrated that the HaloTag-TEV cassette can be used to examine the mechanical function of proteins in their native context by specifically cleaving a single peptide bond. Previous approaches to interfere with the mechanics of proteins in tissues involve the use of non-specific proteases[31], harsh protein extraction protocols[31], or radiation[37], whose non-specific effects are difficult to assess. Our specific titin-severing experiments show that force transduction along the titin filament accounts for up to two-thirds of the stiffness of the skinned mouse psoas muscles measured here (Fig. 2c–e). This magnitude is similar to that inferred from traditional high ionic strength extraction methods[31]. The non-titin-based stiffness is probably originated from other cytoskeletal components, such as microtubules[69], and the extracellular matrix[31]. Structurally, we have observed that TEV-mediated severing of titin leads to sarcomeres that are very sensitive to mechanical loading and unloading cycles (Supplementary Fig. 10), confirming that titin's structural and mechanical support requires a continuous titin polypeptide[70]. In this context, testing the effects induced by titin severing during active contraction will be interesting to understand the coordinated activity of titin and other constituents of the sarcomere.

In summary, we show that the HaloTag-TEV is a non-toxic genetic cassette that can be used to explore and interfere with the mechanical function of proteins from natural sources, while enabling precise fluorescent localization. Indeed, the wide versatility of HaloTag-mediated click chemistry can give rise to biological insights beyond protein mechanics in the system of interest, including, but not limited to, characterization of protein–protein and protein–DNA interactions, and in vivo protein localization[71]. We anticipate that the emergence of highly efficient genome-editing technologies[72] will boost insertion of the HaloTag-TEV cassette in different genes coding for mechanical proteins, and enable screenings for biomechanical functions of proteins. In this regard, the high performance HaloTag-based protein purification can make it possible to examine the mechanics of the many low-abundance proteins that are refractory to recombinant production[73].

## Methods

**Animal experimentation.** All experimental procedures involving animals were done according to CU Institutional Animal Care and Use Committee, the Area de Protección Animal de la Comunidad de Madrid (PROEX 042/18) and the Animal Care and Use Committee of the University Clinic Muenster (A18.016). Mice were euthanized by cervical dislocation and deep-frozen in liquid $N_2$. Tissue biopsies were preserved at −80 °C until use. Serum samples from female WT and homozygous HaloTag-TEV-titin mice were analyzed with Dimension RxL Max integrated chemistry system (Siemens). Basal cardiac function was examined by echocardiography using a 30-MHz linear probe (Vevo 2100, Visualsonics Inc., Canada). Mice were anesthetized in a heating platform using isoflurane, the delivery of which was adjusted so that the heart rate was between 400 and 500 beats per minute. Diastolic function was evaluated by pulsed-wave (PW) Doppler using a 2D apical view, to estimate mitral valve inflow pattern. Blind analysis was performed using the Vevo 2100 Workstation software. Endurance tests were done according to published protocols[74] using a Touchscreen Treadmill (Panlab, Harvard Apparatus) with no further modifications. Mice were trained for 7 days before the actual endurance test.

**Generation of HaloTag-TEV titin knock-in mice.** The HaloTag-TEV cassette was inserted in the 3′ end of titin's exon 225, which codes for the I86 domain (residues 14072–14157, UniProt A2ASS6-1). I86 is located at the end of the I-band of titin (Fig. 1 and Supplementary Fig. 1). The HaloTag-TEV cassette includes a copy of the HaloTag cDNA, which is preceded by a linker coding for a TEV site (Supplementary Note 1). The cassette also includes an enterokinase (EK) site following the HaloTag gene, which we have not used in this report. Our gene targeting strategy is based on homologous recombination, similar to previous modifications of the titin gene, *TTN*[36,75]. We used a Neomycin-resistance (neo) cassette that was modified to include the *TTN* gene region between exon 224 and 234 and the HaloTag-TEV cassette. Embryonic stem (ES) cells were transfected with the linearized targeting vector. Geneticin resistant colonies were screened for homologous recombination by PCR using primers *P1* and *PR2* (left arm), and *P2* and *PR3* (right arm) (Supplementary Fig. 1, Supplementary Table 1). Double PCR-positive ES cells were used for blastocyst injection. Injected blastocysts were transferred to pseudopregnant mice. Chimera mice carrying the targeted allele were identified by PCR, confirming the presence of the left and right arms of the construction (Supplementary Fig. 1). The neo cassette was removed by crossing heterozygous recombinant mice with Flp mice, which express the FRT-Flp recombination system. The offspring was screened by PCR using the $P_{min}$ and $PR_{min}$ primers (Supplementary Fig. 1, Supplementary Table 1), which amplify a segment of the modified intron 225 (size of 175 bp in wild-type mice and 223 bp in knock-in mice). Mice were genotyped by PCR using the same set of primers $P_{min}$ and $PR_{min}$ (Supplementary Fig. 1, Supplementary Table 1).

**Fluorescence microscopy of intact muscle.** For intact muscle visualization, samples from homozygous HaloTag-TEV mice were incubated overnight with 5 μM HaloTag Oregon Green ligand (Promega) in Phosphate-buffered saline (PBS) at 4 °C in a rocking platform, and then fixed in 4% paraformaldehyde for 2 days at 4 °C. After washing two times with PBS, 2-mm muscle slices were clarified using an X-Clarity system (Logos Biosystems). This protocol resulted in homogeneous labeling of muscle fibers, as observed by multiphoton microscopy (Supplementary Movie 1). For multiphoton microscopy, we used a Zeiss LSM780 microscope coupled to an upright Axio Examiner Z1, a water dipping ×20 plan-apochromat NA1 objective, and a Mai Tai DS 800 nm excitation laser (Spectra-Physics). For confocal and STED microscopies, we used a Leica SP8 gSTED 3D system coupled to an inverted DMi6000 microscope, and an HC PL Apo CS2 ×100/1.4 oil objective. We excited at 496 nm and depletion was achieved using a 592 nm laser optimized for xy resolution.

**Light microscopy of isolated myofibrils.** Myofibrils from heterozygous and homozygous mice were isolated according to Linke et al.[76], leading to equivalent results. Briefly, 100 mg of muscle tissue was cut in ~1–2 mm³ fragments and suspended in 500 μL rigor solution (10 mM Tris-HCl buffer pH 7.0, 75 mM KCl, 2 mM MgCl$_2$, and 2 mM EGTA) supplemented with 10 μM leupeptin and 1 mM PMSF. Samples were demembranated by incubation with 1% Triton X-100 for 30 min at 4 °C. After three washes in rigor buffer without detergent, tissues were homogenized using an ultra-turrax disperser (T25, IKA) and kept in ice for 15–30 min. Myofibrils were collected from the top of the suspension. 50–100 μL of the myofibril-containing suspension were diluted three times in rigor buffer supplemented with 10 μM leupeptin. For fluorescence microscopy, we added 10 μM of HaloTag Alexa-488 ligand (Promega) and Alexa Fluor 647 phalloidin (Thermo

Fisher), and incubation proceeded for 30 min at 4 °C followed by three washes with rigor solution. For confocal and spinning disk imaging, myofibrils were deposited on a 35 or 50 mm glass fluorodish (MatTek Corporation) for 15 min, allowing the adhesion of the myofibrils on the surface. Myofibrils were imaged using a Leica TCS SP5 and SP7 confocal microscopes, using either 488 or 649 nm illumination laser with the proper set of filters. Images were analyzed using Leica suited software and Image J.

**Determination of sarcomere length of cardiomyocytes.** Hearts from 2 WT and 3 homozygous HaloTag-TEV mice were used to isolate cardiomyocytes. To that aim, hearts were defrosted in cold relaxing solution (pH 7.0, 7.8 mM ATP, 20 mM imidazole, 4 mM EGTA, 12 mM Mg-propionate, 97.6 mM K-propionate) supplemented with 1:200 Protease Inhibitor Cocktail Set III, EDTA-Free (MERCK Millipore Calbiochem). Heart tissue was disrupted using an IKA T10 Basic Ultra-Turrax at 4 °C. Cardiomyocytes were skinned with 0.5% Triton X-100 diluted in relaxing solution (7 min, on ice). Cells were washed three times in relaxing solution, a drop of the cell suspension was placed on a cover glass under a Zeiss Axiovert 135 inverted microscope as part of a the 1600A Permeabilized Myocyte Test System (Aurora Scientific), and single cells were selected for pickup by two needle tips connected to a piezomotor and force transducer, respectively, aided by Shellac glue. Sarcomeres were visualized using phase-contrast microscopy, a Zeiss ×40 objective (NA 0.60, PH2), and the system's HVSL camera. The slack sarcomere length of the cell in relaxing solution was determined as the length at which there were no compressive or tensile forces acting on the cardiomyocyte. Sarcomere length was measured using the software ASI 901D Hi-Speed Video Sarcomere Length, Version 4.157 (Aurora Scientific).

**Transmission electron microscopy.** Skeletal muscle of 16-week-old homozygous HaloTag-TEV mice was fixed in 4% paraformaldehyde, 15% saturated picric acid in 0.1 M phosphate buffer, pH 7.4, at 4 °C overnight. For conventional ultrastructural analysis, tissue samples were embedded in resin and processed for transmission EM using a VT 1000S Leica vibratome (Mannheim, Germany). To visualize the HaloTag in situ, samples were blocked in 20% normal goat serum (NGS) for 1 h and incubated with the HaloTag® Biotin Ligand (Promega, 200-fold dilution) in PBS supplemented with 5% NGS overnight at 4 °C. The sections were then triple-washed with PBS and incubated with 1.4 nm gold-coupled Nanogold®-Streptavidin (Nanoprobes, Stony Brook, NY, USA) overnight at 4 °C. After extensive washing, all sections were postfixed in 1% glutaraldehyde for 10 min and after rinsing, they were reacted with HQ Silver kit (Nanoprobes) to increase the apparent gold-particle diameter. After treatment with OsO₄, samples were counterstained with uranyl acetate in 70% ethanol, dehydrated and embedded in Durcupan resin (Fluka, Switzerland). Resin blocks were made and ultrathin sections prepared with a Leica Ultracut S (Mannheim, Germany). Sections were adsorbed to glow-discharged Formvarcarbon-coated copper grids. Images were taken using a Zeiss LEO 910 electron microscope equipped with a TRS sharpeye CCD Camera (Troendle, Moorenwies, Germany).

**HaloTag-TEV-titin purification.** Titin molecules were purified using the method implemented by Soteriou et al.[77], adapted to mouse skeletal and cardiac tissue. Briefly, 100–200 mg of muscle tissue in small pieces of ~2 mm³ were suspended in 600–1200 μL of prechilled homogenization buffer (1 mM NaHCO₃ pH 7.0, 50 mM KCl, 5 mM EGTA, and 0.01% NaN₃), supplemented with protease inhibitors (1 mM PMSF, 40 μg mL⁻¹ leupeptin, 20 μM E-64 and 20 μg mL⁻¹ trypsin inhibitor). Tissue was homogenized using an ultra-turrax dispenser (T25, IKA). The resulting suspension was washed four times with 300–600 μL of homogenization buffer. Myofibrils were pelleted by centrifugation at 2000 × g and 4 °C. The pellet was suspended in 1 mL of prechilled extraction buffer (10 mM Imidazole pH 7.0, 900 mM KCl, 2 mM EGTA, 0.01% NaN₃, and 2 mM MgCl₂), supplemented with protease inhibitors (1.5 mM PMSF, 80 μg mL⁻¹ leupeptin, 40 μM E-64 and 40 μg mL⁻¹ trypsin inhibitor), using plastic pellet pestles (Sigma-Aldrich). The extraction was conducted for 5 min on ice, followed by centrifugation at 20,000 × g for 30 min at 4 °C. The supernatant was diluted four times with prechilled precipitation buffer (0.1 mM NaHCO₃ pH 7.0, 0.1 mM EGTA, and 0.01% NaN₃) supplemented with 2 μg mL⁻¹ leupeptin. The solution was incubated for 1 h in ice and centrifuged at 20,000 × g for 30 min at 4 °C to precipitate myosin. The supernatant, rich in titin molecules, was finally diluted with 5 volumes of the same precipitation buffer supplemented with 2 μg mL⁻¹ leupeptin to reach a final concentration of KCl of 0.045 M. After 40 min of incubation on ice, the solution was centrifuged at 10,000 × g for 30 min at 4 °C. The pellet of this last step contains the HaloTag-TEV titin molecules, which were suspended in ~500 μL of 30 mM potassium phosphate buffer pH 7.0, 200 mM KCl and stored at 4 °C.

**TEV-digestion assays.** TEV protease was produced from vector pMHT238Delta[78] or acquired commercially from Thermo Fisher (AcTEV Protease) and used according to the manufacturer's instructions. A cDNA coding for the HaloTag-TEV cassette flanked by domains I86 and I87 was synthesized by Geneart (Supplementary Note 2), and was cloned in the E. coli expression vector pQE80 (Qiagen). Expression of TEV protease and I86-HaloTag-TEV-I87 was induced in BLR (DE3) cells at OD₆₀₀ = 0.6–1.0, using 1 mM IPTG, 3 h at 37 °C, or with 0.4

mM IPTG overnight at 16 °C, respectively. Proteins were purified by Ni-NTA and size-exclusion chromatographies and eluted in 10 mM Hepes, pH 7.2, 150 mM NaCl, 1 mM EDTA, as described[17]. I86-HaloTag-TEV-I87 was stored at 4 °C. TEV was stored at −80 °C after addition of 10% glycerol. Digestion of I86-HaloTag-TEV-I87 by TEV was done in 10 mM Hepes, pH 7.2, 150 mM NaCl, 1 mM EDTA, 10% glycerol, 1 mM DTT. Before SDS-PAGE analysis, samples were incubated with 50 μM HaloTag Alexa488 ligand for 20 min in the dark. For digestion of titin in muscle samples, defrosted muscle tissue was skinned in relaxing buffer to which 0.5% Triton X-100 was added, overnight at 4 °C. After extensive washing and centrifugation in relaxing buffer, samples were incubated in 100 μL relaxing buffer in the presence of 10 μl AcTEV protease (100 units), 7.5 μl TEV buffer 20x, 1.5 μl DTT 0.1 M, 31 μl relaxing buffer for up to 6 h. For fluorescence detection of titin via the HaloTag, we added 10 μM of HaloTag Alexa-488 ligand to a suspension of muscle tissue and incubated for 30 min at 4 °C followed by three washes with relaxing solution. Samples were prepared for protein gel electrophoresis using published protocols[79]. SDS–PAGE was carried out using the Laemmli buffer system in slab gels containing 12.5% polyacrylamide. For titin analysis, 1.8% SDS–PAGE was performed as described[79].

**Muscle fiber mechanics.** Using permeabilized skeletal myofiber bundles dissected from WT mice or animals homozygous for HaloTag-TEV-titin, we measured passive force over the 2.2–3.4 μm sarcomere-length (SL) range. Force measurements were performed according to published protocols[31]. Briefly, deep-frozen tissue was defrosted and skinned overnight in ice-cold low ionic-strength buffer (75 mM KCl, 10 mM Tris, 2 mM MgCl₂, 2 mM EGTA, and 40 μg mL⁻¹ protease inhibitor leupeptin, pH 7.2) supplemented with 0.5% Triton X-100. Under a binocular (Leica, Mannheim, Germany), small bundles of muscle fibers (diameter, 200–300 μm) were dissected and suspended between two mini forceps attached to a piezomotor and a force transducer (Scientific Instruments, Heidelberg, Germany). Force measurements were carried out in relaxing buffer at room temperature. Stretching of the fibers was done stepwise from slack SL in six quick steps. Following each step the fiber was held at a constant length for 10 s to allow for stress relaxation. After the last step-hold, the fiber was released back to slack length. SL was measured by laser diffraction. After a rest time of 10 min, the measurement was repeated. Then, the fiber was treated with recombinant TEV protease and the passive force-SL recordings were repeated at regular intervals. In between measurements, fibers were kept at slack length. For conventional titin extraction experiments, the specimens were treated with buffer containing 0.6 M KCl and 1.0 M KI, following specific titin cleavage by TEV protease. For data analysis in stretch-release experiments, we considered only the peak force levels at the end of each step. For a given experiment, force was normalized to that measured before addition of TEV protease at 3.4 μm SL. Mean data points and SEM from n = 6 experiments were calculated and fit with a 4-term polynomial. An alternative experimental protocol consisted in stretching fiber bundles of WT or homozygous HaloTag-TEV mice (diameter, ~150 μm) in relaxing solution from slack to 3.3 μm sarcomere length only once, and holding them at this length for 40 min. Ten minutes following the stretch, TEV protease was added, while the force level was continuously recorded. After completion of the mechanical experiment, fibers in the stretched state were quick-frozen in liquid nitrogen to be used later for gel electrophoresis and electron microscopy.

**Force spectroscopy by magnetic tweezers.** The experiments were carried out on halo-ligand fluid chambers, prepared according to published protocols[15,18]. Briefly, 3-aminopropyl-trimethoxysilane amine (Sigma-Aldrich) is added to cleaned glass surfaces, followed by glutaraldehyde (Sigma-Aldrich) to crosslink the silane groups and the amine group of halo-ligand (HaloTag amine O4 ligand, Promega) (Fig. 3b). T12-antibody-coated paramagnetic beads were incubated with native Halo-TEV-titin molecules, and then added into the fluid chamber. T12-coated beads were prepared according to Mártonfalvi et al.[20], including some modifications. Briefly, an aliquot of 100 μL carboxylic acid paramagnetic Dynabeads M-270 (Invitrogen) and 10 μL of 5 mg mL⁻¹ of protein A (Sigma-Aldrich), were added to 1 mL of 10 mg mL⁻¹ of N-(3-dimethylaminopropyl)-N′-ethylcabodiimidie hydrochloride (EDC, Sigma-Aldrich) dissolved in 100 mM sodium phosphate, pH 5.0 (2 h at room temperature). Protein A-modified beads were recovered after centrifugation at 3000 × g for 5 min at 4 °C, and washed three times with 1 mL of 200 mM KCl+AB buffer (25 mM Imidazole pH 7.4, 25 mM KCl, 4 mM MgCl₂, 1 mM EGTA). Aliquots of 400 μL of protein A-modified beads were resuspended in 100 mM sodium borate buffer pH 8.2 (three washes). To coat the beads with the anti-titin antibody, 35 μL of 1 mg mL⁻¹ T12 antibody was added to the bead suspension (overnight incubation, 4 °C). Protein A-T12-beads were washed three times with 800 μL of 100 mM sodium borate buffer pH 8.2, and then transferred to 800 μL of 200 mM triethanolamine pH 8.2. One milliliter of bead suspension was washed two times with 200 mM triethanolamine pH 8.2. Beads were then suspended in freshly prepared 50 mM dimethylpimelimidate (DMP) in 200 mM triethanolamine pH 8.2 (45 min at room temperature). Next, the beads were centrifuged and resuspended in 800 μL of 50 mM ethanolamine pH 8.2 (5 min at room temperature). The beads were then suspended in 800 μL of 100 mM sodium borate buffer pH 8.2, followed by two washes with 800 μL of 200 mM KCl+AB buffer. The beads were finally suspended in 400 μL of 200 mM KCl+AB buffer, and could be stored at 4 °C for weeks.

For MT experiments, 100 µL of T12-coated paramagnetic beads were mixed with 400 µL of blocking buffer (20 mM Tris-HCl pH 7.4, 150 mM NaCl, 2 mM MgCl$_2$, 0.01% sodium azide, and 1% BSA). After overnight incubation at 4 °C, the beads were centrifuged and washed three times in 200 mM KCl+AB buffer. Finally, 50 µL of blocked T12-coated beads were incubated with 10–50 µL of purified HaloTag-TEV titin for at least 1 h. Then, the beads were suspended in 200 mM KCl+AB buffer and added to a Halo-ligand functionalized fluid chamber in the absence of magnetic field, as described before[15,18]. The beads were incubated on the surface during 1–5 min to achieve covalent binding between the HaloTag moiety and its ligand. Hundred microliters of 200 mM KCl+AB buffer were added to wash out non-attached paramagnetic beads (the majority, ensuring that most remaining beads have single tethers). During the pulling experiments, the position of the paramagnetic and reference beads was recorded, together with the position of the magnets (MP)[18]. As reference beads, we used 3.57-µm diameter amine bead (Spherotech) to ensure high length working range. For long-term experiments, the buffer was exchanged every ~1–2 h or the fluid chambers were sealed to minimize buffer evaporation. Experimental recordings showing evidence of multimerization (e.g., low unfolding kinetics and low step sizes due to reduced force per molecule) were not included in the analysis[80].

To measure folding probability, we used a multi-pulse force protocol, recruiting Ig domains to the unfolded state at high pulling forces and then relaxing to low forces to allow refolding[15]. We calculated the folding probability by dividing the number of unfolding steps in the final probe pulse by the total Ig domains present in the initial fingerprint pulse. The work ($W$) produced by folding events at each force was calculated considering the change in the end-to-end length of the folding Ig domain ($x$) and the force ($F$) at which the folding events occurs ($W = F \cdot x$)[15,16]. To take into account that the probability of folding is strongly force-dependent, we determined the expected work generated by titin by multiplying the produced work by the folding probability at each force[15].

To estimate the $l_k$ of single-titin tethers, we determined initial extensions before unfolding events at different forces (Δ Length). Our reference force was 1.1 pN, and the difference in length at increasing forces was measured. We estimated $l_k$ by fitting the freely jointed chain model (FJC)[56]:

$$\Delta \text{ Length} = L_C \cdot (\cot h(F \cdot l_k/k_B \cdot T) - k_B \cdot T/(F \cdot l_k)) \\ - L_C \cdot (\cot h(1.1 \cdot l_k/k_B \cdot T) - k_B T/(1.1 \cdot l_k))$$

where $L_C$ is the contour length of the molecule, $F$ is the applied force, $k_B$ is the Boltzmann constant, and $T$ is the absolute temperature.

**Reporting summary**. Further information on research design is available in the Nature Research Reporting Summary linked to this article.

## Data availability
Data and materials are available on reasonable request to the corresponding authors. The source data for Figs. 2D, 3H, 4B and Supplementary Figs. 2 and 3B are provided as a Source Data file, including results from statistical tests if pertinent.

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

## Acknowledgements

This work was supported by the National Institutes of Health grants GM116122 and HL61228 (J.M.F.). J.A.R.P. acknowledges funding from CONICYT grant 11180705. J. A.C. acknowledges funding from the Ministerio de Ciencia e Innovación through grants BIO2014-54768-P, BIO2017-83640-P, and RYC-2014-16604, and the Regional Government of Madrid (S2018/NMT-4443). The CNIC is supported by the Instituto de Salud Carlos III (ISCIII), the Ministerio de Ciencia e Innovación and the Pro CNIC Foundation, and is a Severo Ochoa Center of Excellence (SEV-2015-0505). W.A.L. acknowledges funding from the German Research Foundation (SFB1002, TPA08) and IZKF Munster (Li1/029/20). J.A.C. and W.A.L. acknowledge funding from the European Research Area Network on Cardiovascular Disease through consortium MIN-OTAUR (ISCIII-AC16/00045). Z.M. acknowledges funding from the National Research, Development and Innovation Office (NKFIH) through grants PD116558, FK128956, NVKP-16-1-2016-0017, the Hungarian Academy of Sciences Bolyai grant and an EMBO short term fellowship (ASTF 360-2016). A.F.T. was the recipient of a travel fellowship from the Boehringer Ingelheim Fonds. We thank Consuelo Ibar (Rutgers University) for her help during the confocal measurements, and Dieter Fürst (University of Bonn) for providing the T12 titin antibody. We thank all the members of the Fernandez laboratory for their helpful discussions, and Natalia Vicente (through grant PEJ16/MED/TL-1593 from the Regional Government of Madrid), CNIC's Microscopy, Animal House and Advanced Imaging Facilities, and Marion von Frieling-Salewsky for excellent technical support. Knock-in animals were generated by Thomas Doetschman, Deborah Stead, and Teodora Georgieva from the Bio5 GEMMCore (University of Arizona).

## Author contributions

J.A.R.P., J.A.C., and J.M.F. conceived the project. J.A.R.P., J.A.C., W.L., and J.M.F. designed research. J.A.R.P., A.F.T., E.H.G., and J.A.C. conducted light microscopy experiments. D.V.C. and A.F.T. conducted proteolysis experiments on recombinant proteins. A.F.T. assessed health status of transgenic mice. Y.L. conducted the muscle mechanics experiments and digestion of myofibrils. A.U. completed electron microscopy studies. J.A.R.P., R.T.R., and Z.M. purified titin and did the magnetic-tweezers experiments. J.A.R.P., J.M.F., and J.A.C. drafted the paper with input from all authors.

## Competing interests

The authors declare no competing interests.
