## [Peer Review File · Nature Communications]

Reviewers' Comments:

Reviewer #1:

Remarks to the Author:

Titin is a giant protein in striated muscles that assembles a large macromolecular structure, the sarcomere. Titin is required to provide an elastic linkage between the actin and myosin filament systems, apart from functions in protein complex assembly and signalling. The exact contribution of titin to muscle elasticity has been a subject of controversy, as it was not possible to precisely separate the contributions of sarcomeric elastic links, the cellular cytoskeleton, or the extracellular matrix.

Rivas-Pardo and co-workers have developed an innovative approach to interrogate the elastic functions of titin by both experiments in intact sarcomeres as well as in defined isolated fragments. The genetic engineering of highly specific protease cleavage sites (TEV protease) and attachment sites (Halo-tag) into murine titin allows this study to determine with unprecedented precision the contribution of titin to muscle passive tension in several muscle types. This is a mostly well-written and illustrated manuscript. The novel data reveal that titin contributes only about 40% to muscle passive tension, less than extrapolated by previous studies, and confirms in a very refined way how the stiffness of titin depends on muscle-specific isoforms, with cardiac titin being stiffest.

The second part of the study uses titin isolated from genetically modified mouse muscle and specifically attached via antibody and covalent capture. Again, this is a considerable technical advance and may well be usefully applied to many other mechanically active proteins. The authors conclude that the refolding of single titin domains can generate 41.5 zJ of work.

There are a number of important questions raised by this work.

1. The authors state that unfolding of titin occurs during the lengthening phase of muscle activity. Muscle lengthening occurs due to the antagonist contraction in skeletal muscle or due to the diastolic filling pressure generated by the atrial contraction and the preceding heartbeat in cardiac muscle.

Both are processes of active contraction. The only process in muscle that generates active contraction is unequivocally the actomyosin system that converts chemical energy from ATP into mechanical work. There is no muscle contraction without ATP or after inhibition of myosin ATPase activity.

Therefore, passive lengthening of muscle requires the active, ATP-consuming contraction of the antagonists, or the heart itself by the kinetic energy delivered by the returning venous blood and the contraction of the atria. Unfolding of titin domains therefore requires the action of actomyosin.

The authors conclude, "the work generated by titin folding is of the same order of magnitude as the one generated by actomyosin motors" (page 8, lines 1-2 and ref. 21). While this stored elastic energy can be released during an active contraction when titin is shortening, it is therefore still a zero-sum game: There is not more work done by titin than has previously been performed by the myosin motors. Thermodynamically, it has to be less. Therefore, the only net product of "folding contractions" is heat.

2. A second observation that raises the question of the physiological relevance of this process is the time-scale of the "refolding contraction". Figure 4A shows that the initial roughly 100 seconds of the refolding pulse are dominated by elastic recoil, while folding contraction is observed over the following 1500 seconds (25 minutes). The heart rate of adult mice is about 600 beats per minute, or 10 per second. Many muscle types operate at even higher frequencies. The discrepancy between slow protein folding and fast contraction-relaxation cycles makes a significant contribution of "folding contraction" to the physiological heartbeat or other fast cyclical contraction highly

implausible if not impossible.

Specific further points:

i). It is not clear from which mouse muscle the isolated titin for the single-molecule work in Figures 3 and 4 was isolated.

ii). Following from 2. above, the frequency-domain as well as the force regime under which the measurements are obtained are crucial. In Figure 4B, the folding probability of titin domains goes towards zero at 12 pN. However, over what timescale? Figure 4A suggest over up to 1000 seconds, so outside of physiological significance. As each unfolding would also lead to a relaxation of the titin chain and a subsequent length increment of ~30nm, complete unfolding would also not happen under physiological conditions as sarcomere length is additional constrained by other factors, not least the extracellular matrix and anatomical constraints.

Reviewer #2:

Remarks to the Author:

Rivas-Pardo et al. describe an application of HaloTag and TEV protease to measure mechanical properties of titin protein in more "physiological" conditions compared to previous studies, focusing on the N-terminal third of the protein (titin I-band). Despite a tagged protein is not exactly native as claimed, authors took advantage of model protein expression directly in vivo in mouse and stated that HaloTag-TEV-titin mice are completely healthy. Authors correctly characterized appropriate localization of engineered titin in the muscle fibers exploiting the HaloTag with different types of microscopy, showed that TEV protease can cleave engineered titin, treated myofibres boundles with TEV protease observing a time dependent decrease in the mechanical response of muscle fibers to length changes and performed a detailed analysis of titin I-band response to different force intensities.

The approach is clever; it may be applied more in general to measure mechanical forces of other proteins or domains. At the same time provided data do not seems to go strongly beyond current knowledge of titin function.

Overall I positively evaluate this manuscript, even though some point needs to be clarified/addressed by the authors.

1) There are multiple important contributions of titin in the muscular tissues, such as stiffness, contraction and force production. HaloTag-TEV titin is reported by the authors to have no adverse phenotypes in animal reproduction and fitness with a very general "appear as healthy", "the cassette does not induce any striking phenotype", "mice are fertile" and reproduce at "Mendelian rates". This information together with data of protein localization and simple imaging of sarcomere size are not necessarily accurate enough to assess wild-type equivalent features of engineered titin protein. In my opinion data presented are not sufficient to understand if HaloTag-TEV titin mice heart and muscles have wild type-like strength, elastic properties, resistance to fatigue etc.. A more detailed characterization of the mouse strain with some hearth and muscle tests is required to assess if the tag may affect in some aspects heart and muscle performance, otherwise HaloTag-TEV titin mechanical measurement might not be so "physiological", "native" and interesting as claimed.

2) In Figure 1 the distance between the bands in the doublet is stated as variable due to different contraction statues of sarcomers. It might be useful for a more accurate tissue characterization if authors could measure these characteristics in relaxed and contracted sarcomeres with a comparison to equivalent tissue of wild-type parental mouse strain.

3) Despite is clear that TEV protease can completely cleave the engineered titin "in up to six hours", it is unclear in Figure 2C how much titin has been digested by TEV protease in 30 minutes. How much titin cleavage does cause the 50% drop observed in passive force? A complete TEV protease digestion must be ensured to measure the titin contribution to passive stiffness. The

same applies to Figure 2D. Can authors show a western blot of titin digestion by TEV protease for these experiments and time points?

4) Authors speculate that the activity produced by titin folding can be largely regulated by muscle chaperones, molecular crowding or post-translational modifications. To show the importance and usefulness of the HaloTag-TEV strategy, the actual contribution of some of these players can be measured in manuscript's experimental setting (e.g. role of disulphide-containing domains in the shorter step sizes, S-glutathionation/phosphorylation impact in titin mechanical performance).

5) Can authors describe the posttranslational modifications, not present in the recombinant titin, which affect cooperativity?

6) Supplementary Figure 2. Are the BF and Alexa 488 in the same scale? Please add scale bar to panel with 488.

Reply to reviewers – NCOMMS-19-08301-T

MS title: “A HaloTag-TEV genetic cassette for mechanical phenotyping of native proteins”

We thank both reviewers for their constructive comments, which we have addressed in the revised manuscript resulting in an improved version of the paper. Below, we provide a point-by-point response to their concerns.

Reviewer #1:

Titin is a giant protein in striated muscles that assembles a large macromolecular structure, the sarcomere. Titin is required to provide an elastic linkage between the actin and myosin filament systems, apart from functions in protein complex assembly and signalling. The exact contribution of titin to muscle elasticity has been a subject of controversy, as it was not possible to precisely separate the contributions of sarcomeric elastic links, the cellular cytoskeleton, or the extracellular matrix.

Rivas-Pardo and co-workers have developed an innovative approach to interrogate the elastic functions of titin by both experiments in intact sarcomeres as well as in defined isolated fragments. The genetic engineering of highly specific protease cleavage sites (TEV protease) and attachment sites (Halo-tag) into murine titin allows this study to determine with unprecedented precision the contribution of titin to muscle passive tension in several muscle types. This is a mostly well-written and illustrated manuscript. The novel data reveal that titin contributes only about 40% to muscle passive tension, less than extrapolated by previous studies, and confirms in a very refined way how the stiffness of titin depends on muscle-specific isoforms, with cardiac titin being stiffest.

The second part of the study uses titin isolated from genetically modified mouse muscle and specifically attached via antibody and covalent capture. Again, this is a considerable technical advance and may well be usefully applied to many other mechanically active proteins.

The authors conclude that the refolding of single titin domains can generate 41.5 zJ of work.

R: We thank the reviewer for the positive appreciation of the novelty of our method to examine protein mechanics in general and titin function in particular.

There are a number of important questions raised by this work.

1. The authors state that unfolding of titin occurs during the lengthening phase of muscle activity. Muscle lengthening occurs due to the agonist contraction in skeletal muscle or due to the diastolic filling pressure generated by the atrial contraction and the preceding heartbeat in cardiac muscle.

Both are processes of active contraction. The only process in muscle that generates active contraction is unequivocally the actomyosin system that converts chemical energy from ATP into mechanical work. There is no muscle contraction without ATP or after inhibition of myosin ATPase activity.

Therefore, passive lengthening of muscle requires the active, ATP-consuming contraction of the

antagonists, or the heart itself by the kinetic energy delivered by the returning venous blood and the contraction of the atria. Unfolding of titin domains therefore requires the action of actomyosin.

The authors conclude, “the work generated by titin folding is of the same order of magnitude as the one generated by actomyosin motors” (page 8, lines 1-2 and ref. 21). While this stored elastic energy can be released during an active contraction when titin is shortening, it is therefore still a zero-sum game: There is not more work done by titin than has previously been performed by the myosin motors. Thermodynamically, it has to be less. Therefore, the only net product of “folding contractions” is heat.

R: We agree with the reviewer that ATP-driven mechanochemistry is at the basis of mechanical work generation by striated muscle. Storage and release of elastic energy by titin can make the system more efficient, but quantifying the degree of such optimization (for which kinetics measurements would be needed, see below) falls outside the scope of the current manuscript. We have edited the text accordingly and removed the sentence alluded to by the reviewer.

2. A second observation that raises the question of the physiological relevance of this process is the time-scale of the “refolding contraction”. Figure 4A shows that the initial roughly 100 seconds of the refolding pulse are dominated by elastic recoil, while folding contraction is observed over the following 1500 seconds (25 minutes). The heart rate of adult mice is about 600 beats per minute, or 10 per second. Many muscle types operate at even higher frequencies. The discrepancy between slow protein folding and fast contraction-relaxation cycles makes a significant contribution of “folding contraction” to the physiological heartbeat or other fast cyclical contraction highly implausible if not impossible.

R: We agree that folding kinetics is a key parameter to define contribution of titin folding to active contraction; however, measuring folding kinetics in long titin molecules is far from trivial, for several reasons including low time resolution of magnetic tweezers, the need to drive unfolding/folding reactions to completion, and the difficulties associated with concomitant recoil of random coil regions in titin (PEVK and N2-Bus). This is why in our manuscript we focus on equilibrium properties. Three very recent contributions address some of the issues associated with measuring folding kinetics: (i) development of fast single-molecule setups (Tapia-Rojo et al, PNAS 116, 7873 (2019)), (ii) theoretical models to analyze fast force-quench experiments in short polyproteins (Eckels et al, Cell Reports 27, 1836 (2019)) and (iii) the HaloTag-TEV model reported in our current manuscript. The integration of these three methods to study folding kinetics of titin requires further instrumental and analysis developments, which we feel fall outside the scope of the current manuscript. In the revised version, we have discussed the importance of folding kinetics in the context of muscle contraction (5th paragraph of the discussion).

Specific further points:

i). It is not clear from which mouse muscle the isolated titin for the single-molecule work in Figures 3 and 4 was isolated.

R: We apologize for omitting this information in the previous version of the manuscript. Titin

was isolated from gastrocnemius muscle in the experiments presented in Figures 3 and 4. Legends to both figures have been updated accordingly.

ii). Following from 2. above, the frequency-domain as well as the force regime under which the measurements are obtained are crucial. In Figure 4B, the folding probability of titin domains goes towards zero at 12 pN. However, over what timescale? Figure 4A suggest over up to 1000 seconds, so outside of physiological significance. As each unfolding would also lead to a relaxation of the titin chain and a subsequent length increment of ~30nm, complete unfolding would also not happen under physiological conditions as sarcomere length is additional constrained by other factors, not least the extracellular matrix and anatomical constraints.

R: As pointed out by the reviewer and addressed above, our experiments examine equilibrium folding probabilities, so we used long force pulses in our force protocols. The 3rd paragraph of the revised discussion now stresses the focus in equilibrium properties. We agree with the reviewer that complete titin unfolding is not expected to occur in vivo and have edited the second paragraph of the discussion to stress this fact.

Reviewer #2:

Rivas-Pardo et al. describe an application of HaloTag and TEV protease to measure mechanical properties of titin protein in more “physiological” conditions compared to previous studies, focusing on the N-terminal third of the protein (titin I-band). Despite a tagged protein is not exactly native as claimed, authors took advantage of model protein expression directly in vivo in mouse and stated that HaloTag-TEV-titin mice are completely healthy. Authors correctly characterized appropriate localization of engineered titin in the muscle fibers exploiting the HaloTag with different types of microscopy, showed that TEV protease can cleave engineered titin, treated myofibres bundles with TEV protease observing a time dependent decrease in the mechanical response of muscle fibers to length changes and performed a detailed analysis of titin I-band response to different force intensities.

The approach is clever; it may be applied more in general to measure mechanical forces of other proteins or domains. At the same time provided data do not seems to go strongly beyond current knowledge of titin function.

R: We thank the reviewer for his encouraging comments regarding the potentiality of the HaloTag-TEV cassette for mechanical phenotyping of proteins. We agree with the comment that a tagged protein is not strictly native, and have edited the manuscript accordingly to better describe our method.

Overall I positively evaluate this manuscript, even though some point needs to be clarified/addressed by the authors.

1) There are multiple important contributions of titin in the muscular tissues, such as stiffness, contraction and force production. HaloTag-TEV titin is reported by the authors to have no adverse phenotypes in animal reproduction and fitness with a very general “appear as healthy”, “the cassette does not induce any striking phenotype”, “mice are fertile” and reproduce at “Mendelian rates”. This information together with data of protein localization and simple imaging of sarcomere size are not necessarily accurate enough to assess wild-type equivalent

features of engineered titin protein. In my opinion data presented are not sufficient to understand if HaloTag-TEV titin mice heart and muscles have wild type-like strength, elastic properties, resistance to fatigue etc.. A more detailed characterization of the mouse strain with some heart and muscle tests is required to assess if the tag may affect in some aspects heart and muscle performance, otherwise HaloTag-TEV titin mechanical measurement might not be so “physiological”, “native” and interesting as claimed.

R: Prompted by the reviewer’s comments, we have expanded the phenotypic characterization of the HaloTag-TEV-titin mice to include blood tests of markers of muscle damage and inflammation, body weight, functional cardiac echocardiography, muscle ultrastructure, and muscle mechanical testing. In all cases, the engineered mice show equivalent results to age-matched, wild-type controls. This expanded phenotypic characterization is now included in new Supplementary Figures S2 and S3, and described at the end of the first paragraph of the revised results.

2) In Figure 1 the distance between the bands in the doublet is stated as variable due to different contraction states of sarcomers. It might be useful for a more accurate tissue characterization if authors could measure these characteristics in relaxed and contracted sarcomeres with a comparison to equivalent tissue of wild-type parental mouse strain.

R: We have used phase contrast microscopy to measure sarcomere length in resting cardiomyocytes from HaloTag-TEV-titin and wild-type mice. We have done these measurements in relaxing buffer since it provides a better definition of slack sarcomere length as compared to Ca²⁺-containing activating buffer or rigor buffer, in which actin-myosin interactions strongly influence the sarcomere length, hindering interpretation of results. We find that resting sarcomeres are slightly longer in HaloTag-TEV-titin cardiomyocytes, which is in agreement with the extra length of the two HaloTag-TEV cassettes per sarcomere. Results are presented in new Supplementary Figure S3C and alluded to at the end of the first paragraph of the revised results.

3) Despite it is clear that TEV protease can completely cleave the engineered titin “in up to six hours”, it is unclear in Figure 2C how much titin has been digested by TEV protease in 30 minutes. How much titin cleavage does cause the 50% drop observed in passive force? A complete TEV protease digestion must be ensured to measure the titin contribution to passive stiffness. The same applies to Figure 2D. Can authors show a western blot of titin digestion by TEV protease for these experiments and time points?

R: In our experiments, there is some variability in the time taken by TEV to achieve full digestion of titin, probably reflecting different width of the fibers and/or different degrees of skinning. The value of approximately 50% reduction in passive force was obtained in experiments in which full titin cleavage was confirmed by subsequent SDS-PAGE analysis. In the revised manuscript, we clarify this important point in the main text (3rd paragraph of the section “TEV-mediated...”) and include SDS-PAGE analysis of the fiber used in Figure 2C (new Supplementary Figure S7D). We have also included a representative experiment showing a time-course of TEV-mediated digestion of titin in which full cleavage is achieved in less than 30 min (new Supplementary Figure S8).

4) Authors speculate that the activity produced by titin folding can be largely regulated by muscle chaperones, molecular crowding or post-translational modifications. To show the importance and usefulness of the HaloTag-TEV strategy, the actual contribution of some of these players can be measured in manuscript's experimental setting (e.g. role of disulphide-containing domains in the shorter step sizes, S-glutathionation/phosphorylation impact in titin mechanical performance).

R: We agree with the reviewer that the HaloTag-TEV titin makes it possible to do these highly informative experiments, as we explicitly mention in the revised discussion (4th paragraph), but we believe that accurate characterization of those modulators, which would necessitate in-depth single-molecule analyses, is beyond the scope of the current manuscript.

5) Can authors describe the posttranslational modifications, not present in the recombinant titin, which affect cooperativity?

R: In the 3rd paragraph of the revised version of the discussion, we explicitly describe the posttranslational modifications that are not present in recombinant titin (oxidation and phosphorylation) and cite recent reviews on the topic.

6) Supplementary Figure 2. Are the BF and Alexa 488 in the same scale? Please add scale bar to panel with 488.

R: Yes. We have clarified this point in the legend to the figure.

Reviewers' Comments:

Reviewer #2:

Remarks to the Author:

I would like to acknowledge the authors for their efforts in addressing all my concerns. The manuscript is now suitable for publication.

Reviewer #3:

Remarks to the Author:

Titin is a myofilament located in the sarcomere of striated muscle. The molecule has been extensively studied during the last several decades, with many important contributions made by the authors of this paper. Previous work addressed the contribution of titin to the passive properties of muscle, through mechanical experiments on muscle cells and muscles tissues before and after extracting titin from the sarcomere (using ionizing radiation, mild trypsin treatments and high ionic strength extraction solutions). In the present study a TEV cleavage cassette with a halo tag at its C-terminus was introduced in titin's distal tandem Ig segment in the mouse (the cassette has been inserted between Ig 86 and Ig 87. ~ 15 domains short of the C-terminal end of the elastic region of titin). Although I have many comments (see below), this is an interesting model that opens up new ways of studying titin and this paper will be widely recognized.

The authors first characterized the model with the most important results shown in Figure 1 that focus on the location of the halo-tag in the sarcomere using immunofluorescence and immunoelectron microscopy with a supplemental figure S3a showing transmission electron micrographs. Although this is useful information, it is much more important to know the ultrastructure of the muscle fiber bundles after TEV protease treatment (and not just untreated tissues) in passively stretched muscle and after activation.

1. Does this new model reproduce the earlier work by Podolsky and Horowitz (Nature. 1986 11-17, Cell Biol. 1987;105(5):2217) that showed that in absence of functional titin the A-band translocates towards the Z-disk during activation? Please show the sarcomere ultrastructure after TEV cleavage in tissues that have been passively stretched and tissues that have been maximally calcium activated. Do A-bands translocate away from their normal central location? What else happens to the sarcomere structure?

2. The TEV cleavage site is in the distal tandem Ig segment about 15 domains from the C-terminus of the extensible I-band segment. Since titin domains in this part of the molecule are known to multimerize (J Mol Biol. 2008;384(2):299-312), the question arises what happens to the two ends that flank the cleavage site after TEV cleavage? Could titin domains on opposite sides of the cleavage site interact and establish a path for force transmission between the A-band and Z-disk (at least in some molecules)? This is a possibility since TEV treatments were performed at the slack sarcomere length where the extensible I-band region of titin is in a highly compact state. It would be important to check this (using immuno-labeling techniques that the authors are expert in).

In Figure 2, mechanical experiments are reported on bundles of fibers isolated from the psoas muscle of mice. Demembrated fibers were incubated with TEV protease, then fiber bundles were stretched in their passive state and passive force was measured. As expected, passive force was lower in bundles that had been TEV protease treated. However, the obtained results are very qualitative and it is simply stated that the drop in passive force is 'more than 50%.' The authors should address the following:

3. It is unclear if digestion was complete in the experiments shown in Figure 2D as no force traces are shown that overlap for different treatment durations (i.e., it is unclear if the 120 min treatment resulted in maximal cleavage and a minimal passive force). Extending the treatment duration to beyond 120 min should be used to settle this question.

4. Fig. 2C, inset shows a control curve in which WT muscle was treated for 30 min, causing only a

small drop in force. However, homozygous TEV cassette containing fiber bundles are treated for 120 min (i.e., 4 times longer than your control). Please show controls treated for 120 min (or longer if that were to be required to get a steady-state minimal passive force).

5. Although gels are shown with maximal cleavage, the way these results are described leaves it unsettled whether these gels use lysates from the tissues used for mechanical experiments (this is critical since the digestion duration will depend on many details that vary from fiber bundle to fiber bundle (e.g. the diameter of bundle)).

6. A side-by-side comparison between the present method and one of the methods previously used by the authors to eliminate the contribution of titin to muscle passive tension is required. At complete TEV digestion (and minimal passive force) the authors should show the percent contribution of titin to total passive force as a function of sarcomere length. Please show how this compares to results obtained with trypsin treatment of the skinned muscle or high ionic strength extraction.

7. Sarcomere length measurements in Figure S3 are suspect. Relaxed wt mouse cardiac myocytes should be around 1.8-1.9 μm and not 1.7 μm as shown here. I suggest to double check these measurements.

The single molecule measurements are also interesting. It deserves mentioning that the tethered molecule is still rather complex and that the schematic shown in Fig. 3D is incomplete in that it does not show the N2A element found in all skeletal muscle titins and in a subgroup of the cardiac titins. Thus the tethered segment contains, the proximal tandem Ig segment, the N2B-U_s, the middle tandem Ig segment, the N2A element (with two unique sequences), the PEVK and the distal tandem Ig segment. The small step sizes measured in the histogram could be derived from any of these unique regions.

8. Please add the N2A element in Fig. 3D. Also add the number of Ig domains between the tethers at least in one of the titin's that were studied. For example in the soleus there are likely a total of ~ 70 Ig domains. Also update the numbering of Ig domains (the first domain after the PEVK is not I80),

9. Please discuss the origin of the small step size in greater detail. Have they been seen in your previous work on recombinant proteins? Could they be derived from the N2B-U_s, the N2A element (with two unique sequences), or the PEVK? Or the A-band segment binding back on the tethered segment? Or the tethered segment at short length interacting with the glass surface?

10. What assurances do you have that your measurements are indeed from single molecules? After all the molecule is likely to multimerize near the halo tag and the T12-coated beads could bind multiple molecules.

11. The length of the tethered segment pulled under low force before unfolding needs discussion. For example in the soleus muscle of Fig. 3E stretched by 19.7 nm the initial extension before unfolding is ~ 200 nm (~ 280 for the gastroc). A force of 19.7 pN will extend the native tandem Ig segment to $\sim 90\%$ of its native contour length, or 90% of 70×4 nm, or ~ 250 nm. The PEVK region at 19.7 nm pulling force will extend to about half of its maximal length (assuming the by the authors provides Kuhn segment length), 50% of ~ 580 nm, or 290 nm. This adds up to ~ 640 nm, much more than the 200-300 nm observed. This is a major issue that needs to be addressed. Along similar lines, how can you assume in 3H that the native tandem Ig segment does not extend in the 1 – 87.7 pN range? With the by the authors assumed Kuhn segment length of 20 nm, it can be calculated that they will extend $\sim 20\%$ of their native contour length or ~ 56 nm. You will have to factor this in in 3H and assess whether this could invalidate your conclusion from 3H that you are pulling an average a single molecule. Your own earlier work on PEVK molecules suggests a Kuhn segment length of the PEVK of ~ 2 nm. You are currently at 1 nm and with the correction for tandem segment extension the final value will be less than 1 nm. You might be pulling doublets?

12. The refolding data in figure 4 and their implication that titin performs mechanical work during shortening are interesting. However, their physiological significance will depend on how many domains unfold during physiological muscle loading. The authors state that physiological force range goes up to 10 pN (second page of discussion), but they have published earlier that the

physiological range in the heart extends to only 5 pN (Nature. 2002;418(6901):998-1002, Fig. 4) and other recent studies on skeletal muscle suggest that physiological forces might actually be below 5 pN (Elife. 2018 Dec 19;7, Fig. 3C and 8D). Thus it seems likely that the fraction of I-band domains that are recruited to the unfolded state is small under physiological conditions. (As a side note, the existence of isoforms of titin with different spring composition makes most sense if that is the case.) Please more realistically discuss the physiological implications of your refolding data.

Minor: in your title you refer to 'native tissues'. Why was 'native' added? Are there non-native tissues that the reader might be confused about if you were to leave out 'native'?

Minor: Please add to the acknowledgements by whom and where this interesting model was made.

Reply to reviewers NCOMMS-19-08301A

Reviewer #2:

I would like to acknowledge the authors for their efforts in addressing all my concerns. The manuscript is now suitable for publication.

R: We thank the reviewer for their contribution leading to an improved version of the manuscript.

Reviewer #3

Titin is a myofilament located in the sarcomere of striated muscle. The molecule has been extensively studied during the last several decades, with many important contributions made by the authors of this paper. Previous work addressed the contribution of titin to the passive properties of muscle, through mechanical experiments on muscle cells and muscles tissues before and after extracting titin from the sarcomere (using ionizing radiation, mild trypsin treatments and high ionic strength extraction solutions). In the present study a TEV cleavage cassette with a halo tag at its C-terminus was introduced in titin's distal tandem Ig segment in the mouse (the cassette has been inserted between Ig 86 and Ig 87. ~ 15 domains short of the C-terminal end of the elastic region of titin). Although I have many comments (see below), this is an interesting model that opens up new ways of studying titin and this paper will be widely recognized.

R: We thank the reviewer for recognizing the value and the potential of our approach to study titin, and for their insightful feedback and comments, which we have addressed to improve the manuscript.

The authors first characterized the model with the most important results shown in Figure 1 that focus on the location of the halo-tag in the sarcomere using immunofluorescence and immunoelectron microscopy with a supplemental figure S3a showing transmission electron micrographs. Although this is useful information, it is much more important to know the ultrastructure of the muscle fiber bundles after TEV protease treatment (and not just untreated tissues) in passively stretched muscle and after activation.

R: Following the recommendation of the reviewer, we have expanded the ultrastructural characterization of TEV-treated muscle, as detailed below.

1. Does this new model reproduce the earlier work by Podolsky and Horowitz (Nature. 1986 11-17, Cell Biol. 1987;105(5):2217) that showed that in absence of functional titin the A-band translocates towards the Z-disk during activation? Please show the sarcomere ultrastructure after TEV cleavage in tissues that have been passively stretched and tissues that have been maximally calcium activated. Do A-bands translocate away from their normal central location? What else happens to the sarcomere structure?

R: these are very interesting questions about the biology of titin. In the revised version of the manuscript, we now include EM data showing massive loss of structure in TEV-treated HaloTag-TEV titin sarcomeres that have been subject to mechanical load (new Supplementary Figure S10). These results go in the same direction as the data in the Podolsky and Horowitz paper. Although very appealing, we believe that studying the effect of TEV-severing during active contraction falls outside the scope of the manuscript, in which we have focused on the canonical function of titin as a main contributor to sarcomere passive elasticity. To accommodate the addition of the new data, we have modified the results section (3rd and 4th paragraphs of section "TEV-mediated polypeptide severing hinders titin mechanical function"). In the text, we also discuss the new data referencing the paper by Podolsky and Horowitz, and acknowledge that using the HaloTag-TEV titin model can be very interesting to understand titin structural

mechanical support in the context of active muscle contraction, as suggested by the reviewer (end of second to last paragraph of the discussion).

2. The TEV cleavage site is in the distal tandem Ig segment about 15 domains from the C-terminus of the extensible I-band segment. Since titin domains in this part of the molecule are known to multimerize (J Mol Biol. 2008;384(2):299-312), the question arises what happens to the two ends that flank the cleavage site after TEV cleavage? Could titin domains on opposite sides of the cleavage site interact and establish a path for force transmission between the A-band and Z-disk (at least in some molecules)? This is a possibility since TEV treatments were performed at the slack sarcomere length where the extensible I-band region of titin is in a highly compact state. It would be important to check this (using immuno-labeling techniques that the authors are expert in).

R: We thank the reviewer for this insight, which is very pertinent for the function of titin. To examine mechanical contribution of multimerization, we have conducted additional TEV-digestion/mechanical testing experiments at high sarcomere lengths (3.3 microns), in which multimerization is minimized (new Figure 2E). In agreement with the multimerization hypothesis, in these experiments the estimated contribution of titin to passive stiffness is higher than in the experiments where TEV protease is added to slack fibers. We now discuss in the text the possibility that multimerization enables some force transduction through titin molecules even when TEV digestion is complete (last paragraph of section “TEV-mediated polypeptide severing hinders titin mechanical function”). We have also modified the legend to Figure 2 and the section “Muscle fiber mechanics” of the methods section to describe the differences between both mechanical protocols.

In Figure 2, mechanical experiments are reported on bundles of fibers isolated from the psoas muscle of mice. Demembrated fibers were incubated with TEV protease, then fiber bundles were stretched in their passive state and passive force was measured. As expected, passive force was lower in bundles that had been TEV protease treated. However, the obtained results are very qualitative and it is simply stated that the drop in passive force is ‘more than 50%.’ The authors should address the following:

3. It is unclear if digestion was complete in the experiments shown in Figure 2D as no force traces are shown that overlap for different treatment durations (i.e., it is unclear if the 120 min treatment resulted in maximal cleavage and a minimal passive force). Extending the treatment duration to beyond 120 min should be used to settle this question.

R: As pointed out by the reviewer in point #5 below, the time for full digestion depends on the specific sample tested probably due to its diameter or degree of permeabilization. In the revised text (3rd paragraph of section “TEV-mediated polypeptide severing hinders titin mechanical function” of the results), we now explain that, in order to quantify force drops induced by TEV, we only considered fibers in which completeness of TEV digestion could be verified by SDS-PAGE (this info was in the legend to Supplementary Figure S8 in the previous version of the manuscript). In the classical stretch-release experiments (Figure 2D), a 50% drop in passive force was consistently obtained, so we have reviewed the manuscript for self-consistency.

4. Fig. 2C, inset shows a control curve in which WT muscle was treated for 30 min, causing only a small drop in force. However, homozygous TEV cassette containing fiber bundles are treated for 120 min (i.e., 4 times longer than your control). Please show controls treated for 120 min (or longer if that were to be required to get a steady-state minimal passive force).

R: In the revised manuscript we now provide WT control data at 120 min (new data in Figure 2D).

5. Although gels are shown with maximal cleavage, the way these results are described leaves it unsettled whether these gels use lysates from the tissues used for mechanical experiments (this is critical since the digestion duration will depend on many details that vary from fiber bundle to fiber bundle (e.g. the diameter of bundle)).

R: see response to point #3

6. A side-by-side comparison between the present method and one of the methods previously used by the authors to eliminate the contribution of titin to muscle passive tension is required. At complete TEV digestion (and minimal passive force) the authors should show the percent contribution of titin to total passive force as a function of sarcomere length. Please show how this compares to results obtained with trypsin treatment of the skinned muscle or high ionic strength extraction.

R: We have done the interesting side-by-side comparison suggested by the reviewer. In the revised version of the manuscript, we show that adding high ionic strength buffer to extract titin from TEV-treated fibers further decreases passive force at all sarcomere lengths (revised Figure 2D; commented in the second to last paragraph of the section “TEV-mediated polypeptide severing hinders titin mechanical function” of the results). This result together with the TEV-digestion experiments at high sarcomere lengths (see response to point #2 above) suggest that there could be some residual force transduction due to titin multimerization. Indeed, the contribution of titin to passive stiffness in the TEV-digestion experiments at high sarcomere lengths is very similar to values reported using traditional extraction methods, as discussed in the second to last paragraph of the discussion of the revised manuscript. Experimental details have been included in the methods section.

7. Sarcomere length measurements in Figure S3 are suspect. Relaxed wt mouse cardiac myocytes should be around 1.8-1.9 μm and not 1.7 μm as shown here. I suggest to double check these measurements.

R: We thank the reviewer for his/her expert advice. Most probably our previous results were affected by incomplete Ca^{2+} removal/skinning leading to residual contraction. In the revised version, we report a new set of data in which we measure a slack sarcomere length of $\sim 1.85 \mu\text{m}$ for both WT and H/H mice (new Figure S3C), supporting that the insertion of the HaloTag-TEV cassette in titin does not affect sarcomere structure. We have adapted the text accordingly (1st paragraph of the results section and the pertinent section of the methods).

The single molecule measurements are also interesting. It deserves mentioning that the tethered molecule is still rather complex and that the schematic shown in Fig. 3D is incomplete in that it does not show the N2A element found in all skeletal muscle titins and in a subgroup of the cardiac titins. Thus the tethered segment contains, the proximal tandem Ig segment, the N2B-Us, the middle tandem Ig segment, the N2A element (with two unique sequences), the PEVK and the distal tandem Ig segment. The small step sizes measured in the histogram could be derived from any of these unique regions.

8. Please add the N2A element in Fig. 3D. Also add the number of Ig domains between the tethers at least in one of the titin's that were studied. For example in the soleus there are likely a total of ~ 70 Ig domains. Also update the numbering of Ig domains (the first domain after the PEVK is not I80),

R: we have updated Fig. 3D to include the reviewer's suggestions, and have also modified accordingly titin schematics in Figure 1A, 2A and 3A for self-consistency. To illustrate the structural complexity of titin we have chosen to represent the N2BA isoform, which is well annotated in uniprot (A2ASS6-1). The figure shows that there are ~75 Ig domains in the tether of the N2BA titin isoform.

9. Please discuss the origin of the small step size in greater detail. Have they been seen in your previous work on recombinant proteins? Could they be derived from the N2B-U_s, the N2A element (with two unique sequences), or the PEVK? Or the A-band segment binding back on the tethered segment? Or the tethered segment at short length interacting with the glass surface?

R: we have seen small step sizes in our previous work with recombinant proteins: disulfide-containing domains (refs. 55-56. in the text), unfolding proceeding through intermediates (ref. 57 in the text) and unfolding/folding transitions of molten globule states (ref 60 in the text). However, the reviewer is right that the small steps could originate also from the N2B-U_s, the N2A elements and the PEVK. It is highly unlikely though that the population of short steps is caused by spurious interactions such as A-band binding or surface interactions because in those cases we would not expect a well-defined step size. To account for all these possibilities, we have expanded our discussion on the origin of the small steps (2nd paragraph of section "The HaloTag-TEV cassette enables directed manipulation of single titin molecules").

10. What assurances do you have that your measurements are indeed from single molecules? After all the molecule is likely to multimerize near the halo tag and the T12-coated beads could bind multiple molecules.

R: Tethers composed of several polyproteins have mechanical fingerprints that make them easy to detect. For instance, if a dimer is formed in a geometry such that domain unfolding is out-of-register, all unfolding steps are split in two (for instance, 25 nm steps can be split in 10 +15 nm steps). If domain unfolding is in register, the step size is preserved, but the force experienced by the domains halves (Sarkar, A. et al. The mechanical fingerprint of a parallel polyprotein dimer. *Biophys J* 92, L36-38. (2007)), which leads to noticeable changes in unfolding kinetics and step size especially at low forces. Traces showing evidence of multimerization in our experiments were discarded. A reassuring observation is that measured step sizes match the ones obtained with single-recombinant polyproteins in previous reports (Rivas-Pardo, J. A. et al. Work Done by Titin Protein Folding Assists Muscle Contraction. *Cell reports* 14, 1339-1347. (2016)). In addition, we work under conditions in which most beads do not have any tether, so based on Poisson statistics we can be confident that the majority of beads analyzed have single-molecule tethers. Hence, both Poisson statistics and the absence of multimer mechanical fingerprints ensure that our results do not originate from titin multimerization. In the revised methods section of the manuscript, we include information on how we can be confident that our measurements are done on single titin monomers (second paragraph of section "Force spectroscopy by magnetic-tweezers").

11. The length of the tethered segment pulled under low force before unfolding needs discussion. For example in the soleus muscle of Fig. 3E stretched by 19.7 nm the initial extension before unfolding is ~200 nm (~280 for the gastroc). A force of 19.7 pN will extend the native tandem Ig segment to ~90% of its native contour length, or 90% of 70 x 4 nm, or ~ 250 nm. The PEVK region at 19.7 nm pulling force will extend to about half of its maximal length (assuming the by the authors provides Kuhn segment length), 50% of ~580 nm, or 290 nm. This adds up to ~640 nm, much more than the 200-300 nm observed. This is a major issue that needs to be addressed.

R: We agree with the reviewer that analysis of initial length can provide extra reassurance that our experiments are probing single I-band titin molecules. As elaborated below, these calculations do not contradict our interpretation of the results. However, we prefer not to include them in the paper because of the assumptions that have to be made regarding the exact number of Ig domains and the length of the PEVK in the specific titin isoforms probed, and the Kuhn lengths of native titin segments.

It is important to stress though that due to the experimental configuration of magnetic tweezers, the initial force is never zero (this is different from AFM, which has been traditionally used to characterize the mechanics of titin at high forces). In the experiments shown in Fig 3E, the initial forces are: cardiac, 1.7 pN ; gastrocnemius, 4.2 pN; soleus, 4.2 pN (now this information is provided in the legend to Figure 3E). Hence, the initial extensions in the panels in Fig. 3E correspond to entropic adjustment from those initial forces to a final force of 19.7 pN. Considering the composition of I-band titin suggested by the reviewer and canonical Kuhn lengths (20 nm for tandem Ig domains and 1.82 nm for PEVK) [Ref. 13 in the text], the Freely Jointed Chain (FJC) model of polymer elasticity estimates that the tandem Ig domains ($n=70$) should extend by 11.2 nm (4% their contour length) and that the PEVK should extend by 214.6 nm (37% its contour length), for a total increase in length of 226 nm. This theoretical value is very similar to the 200-300 nm experimental initial lengths of the traces in Fig. 3E.

To further assess the agreement between theoretical and experimental initial lengths and based on the same mechanical parameters as above, we have also estimated the initial length when taking gastrocnemius titin from 1.1pN to several forces, for which we have experimental data from 7 molecules (Figure 3H). Results show a good agreement between experimental data and the predictions from the FJC model (see figure below).

Overall, the estimates of initial lengths do not challenge our interpretation of the single-molecule results. We have edited the manuscript to stress the fact that the initial force in magnetic tweezers is never zero (beginning of 2nd paragraph of section “The HaloTag-TEV cassette enables directed manipulation of single titin molecules” and legend to Figure 3E).

Along similar lines, how can you assume in 3H that the native tandem Ig segment does not extend in the 1 – 87.7 pN range? With the by the authors assumed Kuhn segment length of 20 nm, it can be calculated that they will extend ~20% of their native contour length or ~56 nm. You will have to factor this in in 3H and assess whether this could invalidate your conclusion from 3H that you are pulling an average a single molecule. Your own earlier work on PEVK molecules suggests a Kuhn segment length of the PEVK of ~ 2nm. You are currently at 1 nm and with the correction for tandem segment extension the final value will be less than 1 nm. You might be pulling doublets?

R: The analysis in Figure 3H aims to test whether there are low-Kuhn-length (random coil) segments in the probed titin molecules, to support specific pulling of the I-band (and not the A-band, which lacks random coil regions). To avoid any preconception about the architecture of titin, we fitted the data to a single-term FJC model (equation at the end of the methods section). Hence, our analyses do not assume that the tandem Ig segment is inextensible. Instead, the results from the FJC fit are a convolution of all the structures present in the tethered titin. The fact that we obtain a low Kuhn length shows that the contribution of random coil regions is high, strongly supporting that we are specifically probing the I-band of titin. Considering our simplified model though, we agree with the reviewer that the obtained 1-nm value represents an upper limit of the PEVK’s Kuhn length, which is however in the range determined before using recombinant fragments (0.8 – 5 nm) [ref. 13 in the text]. Interestingly, posttranslational modifications that can be present in native titin molecules can further reduce the Kuhn length of the PEVK region [Hidalgo and Granzier. *Circ Res* 105, 631-638 (2009)]. We have modified the manuscript to explain better our approach and to discuss the value of Kuhn length obtained in the context of data available in the literature (last paragraph of section “The HaloTag-TEV cassette enables directed manipulation of single titin molecules”).

Regarding the reviewer’s comment about doublets, it is well known that the persistence/Kuhn lengths of parallel dimers halves with respect to true single-molecule tethers (Sarkar, A. et al. The mechanical fingerprint of a parallel polyprotein dimer. *Biophys J* 92, L36-38. (2007)). Unfortunately, under our experimental conditions, it is not possible to assess the presence of doublets from analyses of initial extension because both scenarios would lead to very similar results according to FJC predictions (see figure below). However, we are confident that we are not pulling from doublets from the arguments given in the response to point #10, which are based on much more sensitive unfolding step size and unfolding kinetics.

12. The refolding data in figure 4 and their implication that titin performs mechanical work during shortening are interesting. However, their physiological significance will depend on how many domains unfold during physiological muscle loading. The authors state that physiological force range goes up to 10 pN (second page of discussion), but they have published earlier that the physiological range in the heart extends to only 5 pN (Nature. 2002;418(6901):998-1002, Fig. 4) and other recent studies on skeletal muscle suggest that physiological forces might actually be below 5 pN (Elife. 2018 Dec 19;7, Fig. 3C and 8D). Thus it seems likely that the fraction of I-band domains that are recruited to the unfolded state is small under physiological conditions. (As a side note, the existence of isoforms of titin with different spring composition makes most sense if that is the case.) Please more realistically discuss the physiological implications of your refolding data.

R: We agree with the reviewer that the significance of titin domains performing mechanical work depends on the number of domains unfolded at physiological forces. However, estimates of physiological forces across titin are based on assumptions highly challenging to verify experimentally (exact molecular composition of titin molecules, native persistence lengths, density of titin molecules in myocytes, etc). Although most estimates agree that physiological forces should not exceed 10 pN (papers alluded by the reviewer, and also ref. 21 in the text), actual values have not been measured experimentally. Hence, following the request of the reviewer, we have removed the word “physiological” throughout the manuscript when it was not needed, providing instead actual force values. In addition, we have included a discussion about the range of physiological forces (end of 3rd paragraph of the discussion).

Minor: in your title you refer to ‘native tissues’. Why was ‘native’ added? Are there non-native tissues that the reader might be confused about if you were to leave out ‘native’?

R: We agree with the reviewer that we don’t need the word “native” and that juxtaposition with recombinant proteins is evident. Hence, we have removed the word

“native” from the title. We have also reviewed the use of the word native throughout the text and edited when appropriate.

Minor: Please add to the acknowledgements by whom and where this interesting model was made.

R: we have moved the info from the Methods to the Acknowledgements. In addition, we now include the specific staff involved in the generation of the model.

Reviewers' Comments:

Reviewer #3:

Remarks to the Author:

The authors have responded to all comments in an effective and expert fashion. I have no further comments and congratulate the authors with their interesting animal model and research findings.